# Heg1 and Ccm1/2 proteins control endocardial mechanosensitivity during zebrafish valvulogenesis

Stefan Donat[1,2†], Marta Lourenço[1†], Alessio Paolini[1†], Cécile Otten[1], Marc Renz[1], Salim Abdelilah-Seyfried[1,2]*

[1]Institute of Biochemistry and Biology, Potsdam University, Potsdam, Germany; [2]Institute of Molecular Biology, Hannover Medical School, Hannover, Germany

**Abstract** Endothelial cells respond to different levels of fluid shear stress through adaptations of their mechanosensitivity. Currently, we lack a good understanding of how this contributes to sculpting of the cardiovascular system. Cerebral cavernous malformation (CCM) is an inherited vascular disease that occurs when a second somatic mutation causes a loss of CCM1/KRIT1, CCM2, or CCM3 proteins. Here, we demonstrate that zebrafish Krit1 regulates the formation of cardiac valves. Expression of *heg1*, which encodes a binding partner of Krit1, is positively regulated by blood-flow. In turn, Heg1 stabilizes levels of Krit1 protein, and both Heg1 and Krit1 dampen expression levels of *klf2a*, a major mechanosensitive gene. Conversely, loss of Krit1 results in increased expression of *klf2a* and *notch1b* throughout the endocardium and prevents cardiac valve leaflet formation. Hence, the correct balance of blood-flow-dependent induction and Krit1 protein-mediated repression of *klf2a* and *notch1b* ultimately shapes cardiac valve leaflet morphology.

DOI: https://doi.org/10.7554/eLife.28939.001

*For correspondence:
salim.seyfried@uni-potsdam.de

†These authors contributed equally to this work

**Competing interests:** The authors declare that no competing interests exist.

## Introduction

Biophysical forces including shear stress caused by blood-flow trigger activation of mechanotransduction pathways within endothelial cells (ECs), a process which causes cellular changes that contribute to the sculpting of vascular networks and of the heart (*Baeyens and Schwartz, 2016*; *Haack and Abdelilah-Seyfried, 2016*). Currently, our knowledge is incomplete regarding the modes by which the sensitivity of different vascular beds is attuned to the constantly changing stimulus of blood-flow. One well-studied example of a blood-flow-sensitive developmental process involves remodeling of endocardial cushions into cardiac valve leaflets in the zebrafish embryo (*Beis et al., 2005*; *Pestel et al., 2016*; *Scherz et al., 2008*; *Steed et al., 2016*). Prior to the formation of cardiac cushions, the oscillatory pattern of blood-flow induces expression of the mechanotransduction pathway components zinc-finger transcription factors Krüppel like factors 2a (*klf2a*) (*Vermot et al., 2009*) and *klf2b* (*Renz et al., 2015*). *KLF*s are important blood-flow-responsive genes within ECs that are strongly activated within regions of high shear stress in tissue culture and various *in vivo* models (*Dekker et al., 2002, 2005*; *Groenendijk et al., 2004, 2005*; *Lee et al., 2006*; *Novodvorsky and Chico, 2014*; *Parmar et al., 2005*; *Zhou et al., 2016*). As zebrafish cardiac atrioventricular (AV) cushions are remodeled into AV valve leaflets, endocardial *klf2a* is highly expressed on the luminal side of the developing valve leaflet, which is exposed to blood-flow, whereas its expression is low on the abluminal side of the leaflet (*Steed et al., 2016*). Zebrafish *klf2a* mutants exhibit defective cardiac valve leaflets (*Steed et al., 2016*) and a loss of murine *Klf2* also results in atrioventricular cushion defects (*Chiplunkar et al., 2013*). One of the targets of *klf2a* within cardiac cushions is *notch1b* (*Vermot et al., 2009*). The precise spatiotemporal regulation of *Klf2* expression within cardiac valve

leaflets indicates that fine-tuning the response to blood-flow has a conserved role in shaping functional leaflets during cardiac morphogenesis.

The control of *Klf2* expression has been linked to the cerebral cavernous malformations protein complex (*Maddaluno et al., 2013*; *Renz et al., 2015*; *Zhou et al., 2015*, *2016*). Within this complex, KRIT1 (also known as CCM1) and the transmembrane protein HEG1 (also known as Heart of glass) physically interact with each other. This interaction results in stabilization of both proteins at EC junctions (*Gingras et al., 2012*; *Kleaveland et al., 2009*). The loss of HEG1, KRIT1, CCM2, or PDCD10 (also referred to as CCM3) causes severe cardiovascular and cardiac developmental defects (*Kleaveland et al., 2009*; *Mably et al., 2003*; *Mably et al., 2006*; *Whitehead et al., 2009*; *Yoruk et al., 2012*; *Zheng et al., 2010*). However, functional studies in zebrafish suggest that Ccm3 functions in a manner that is separate from Krit1 and Ccm2 (*Yoruk et al., 2012*). The name CCM refers to pathologies with familial inheritance that have been attributed to mutations in human *KRIT1*, *CCM2*, or *PDCD10*. Affected individuals exhibit morphological malformations of low-perfused venous endothelial beds of the neuro-vasculature that can result in dangerous cerebral bleeding [reviewed in *Riant et al., 2010*]. Similarly, the conditional endothelial-specific loss of *Krit1*, *Ccm2*, or *Pdcd10* in postnatal mice causes CCM lesions (*Boulday et al., 2011*; *Chan et al., 2011*). Strikingly, postnatal loss of *Heg1* in mice does not cause CCM lesions and familial forms of the CCM pathology have never been associated with mutations in *HEG1* (*Zheng et al., 2014*). These findings suggest a developmental role for HEG1-CCM signaling which differs from postnatal functions of the other CCM proteins.

Functional studies in zebrafish *ccm* and murine *Ccm* mutants have linked cardiovascular developmental defects to elevated expression levels of *Klf2* (*Renz et al., 2015*; *Zhou et al., 2015*; *2016*). Two major pathways have been implicated in regulation of *KLF2* by the CCM proteins KRIT1 and CCM2. First, the core adaptor protein KRIT1 and its associated β1-Integrin binding protein 1 (ICAP1) suppress β1-Integrin signaling in human umbilical vein endothelial cells (HUVECs) (*Faurobert et al., 2013*). A loss either of CCM proteins or of ICAP1 enhances Integrin signaling and this causes strong upregulation of *KLF2* expression in HUVECs, which is suppressed by the depletion of β1-Integrin (*Renz et al., 2015*). Similarly in zebrafish, *ccm* mutant cardiovascular defects are suppressed by a knockdown of β1-Integrin (*Renz et al., 2015*). Secondly, the binding of CCM2 to MEKK3 (also known as MAP3K3) (*Cullere et al., 2015*; *Uhlik et al., 2003*; *Zhou et al., 2015*) inhibits the MEKK3-MEK5-ERK5 mechanotransduction pathway, which controls expression levels of murine *Klf2* (*Zhou et al., 2015*). These findings point to the importance of CCM proteins in regulating the physiological responsiveness of ECs to blood-flow. In support of such a physiological role, CCM proteins antagonize activation of β1-Integrin and thus interfere with the orientation of endothelial cells in response to shear stress (*Macek Jilkova et al., 2014*).

The developmental connection of Ccm proteins with the regulation of *klf2a* expression led us to investigate whether expression of zebrafish *heg1* and *krit1* within the vasculature is controlled in a mechanosensitive manner. We find that expression of *heg1* is positively regulated in response to blood-flow and *klf2a/b*. We also explored the role of Heg1 and Krit1 in Klf2-dependent endothelial mechanotransduction during the formation of cardiac valve leaflets. Here, we show that Krit1 dampens levels of *klf2a* and *notch1b* expression. In tune with this finding, loss of Krit1 results in strong expression of *klf2a* and *notch1b* within the endocardium and prevents formation of an abluminal population of valve leaflet cells from endocardial cushions. Our findings reveal an important developmental role of Ccm proteins in fine-tuning the sensitivity of ECs in response to the strength of the biomechanical stimulus of blood-flow during valvulogenesis.

## Results

### Expression levels of *heg1* mRNA are positively regulated by blood-flow and Klf2a/b

To test whether the regulation of *heg1* or *krit1* mRNAs responds to changes in blood-flow, we measured their expression levels using RT-qPCR in *troponin T type 2a* (*tnnt2a*) morphants that have a non-contractile heart and thus lack blood-flow (*Sehnert et al., 2002*). Under this condition, mRNA levels of *heg1* but not of *krit1* were significantly lower than in wild-type (WT) embryos at 54 hr post fertilization (hpf) (*Figure 1A*). Similarly, antisense oligonucleotide morpholino (MO)-mediated

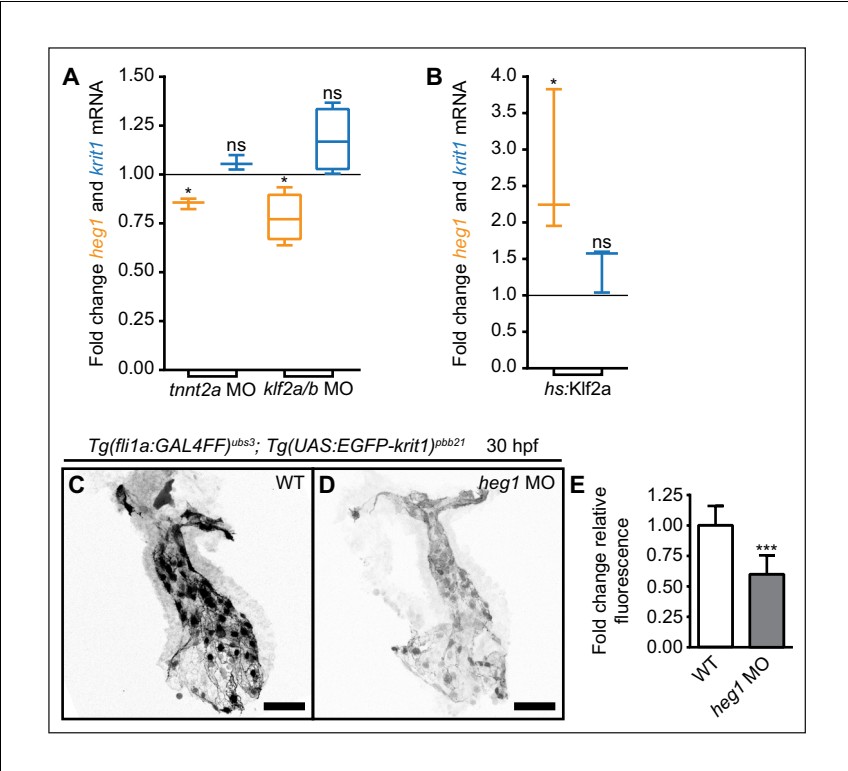

**Figure 1.** Mechanosensitive expression of *heg1* mRNA and stabilization of Krit1 protein levels by *heg1*. (A) RT-qPCR quantifications of *heg1* and *krit1* mRNA levels at 54 hpf in morphant embryos lacking blood-flow [*troponin T type 2a* (*tnnt2a*)], or Krüppel like factors 2a and 2b (*klf2a/b*) compared with wild-type expression levels. (B) RT-qPCR quantification showing that heat-shock-induced overexpression of *klf2a* causes significant upregulation of *heg1* but not of *krit1* expression at 54 hpf. (C–E) Morpholino-mediated knockdown of Heg1 significantly reduces endocardial EGFP-Krit1 protein levels of 30 hpf *Tg(fli:GAL4FF)ubs3*; *Tg(UAS:EGFP-Krit1)pbb21* embryos. Scale bars are 50 μm. Mean values ± SEM of three (*tnnt2a* MO, *hs:klf2a*, *heg1* MO) or four (*klf2a/b* MO) individual experiments are shown. Ratio paired (A, B) or unpaired (E) t-test was used to compare each condition with controls (ns: not significant, *p<0.05, ***p<0.001).

DOI: https://doi.org/10.7554/eLife.28939.002

The following figure supplements are available for figure 1:

**Figure supplement 1.** Whole-mount *in situ* hybridization of *heg1* cardiac expression.

DOI: https://doi.org/10.7554/eLife.28939.003

**Figure supplement 2.** Stabilization of Krit1 protein levels by *heg1* in the trunk vasculature.

DOI: https://doi.org/10.7554/eLife.28939.004

knockdown of Klf2a/b caused significant downregulation of *heg1* mRNA. In comparison, the expression levels of *krit1* mRNA were not significantly altered (*Figure 1A*). To test whether *heg1* is positively regulated by Klf2a, we generated transgenic lines of zebrafish for heat-shock-mediated induction of *klf2a* [*Tg(hsp70l:klf2a_IRES_EGFP)pbb22*]. We found that overexpression of *klf2a* upon heat-shock at 48–50 hpf led to a significant upregulation of *heg1* mRNA expression by 54–55 hpf (*Figure 1B*). However, the treatment did not significantly upregulate levels of *krit1* mRNA.

It has previously been shown that *heg1* is expressed within endocardium (*Mably et al., 2003*). To determine the regional expression of *heg1* mRNA within the endocardium, we performed whole-mount *in situ* hybridizations in WT and *tnnt2a* morphants. Whereas *heg1* was initially expressed throughout the entire endocardium at 30 hpf, by 48 hpf, expression became more restricted to cells exposed to a high fluid shear stress including the atrioventricular canal (AVC) endocardium, and this was even more striking at 72 hpf [*Figure 1—figure supplement 1A–C*; (*Münch et al., 2017*)]. In tune with flow-dependent regulation within the endocardium, expression of *heg1* mRNA was reduced in *tnnt2a* morphants at 48 hpf (*Figure 1—figure supplement 1D*).

As Heg1 and Krit1 proteins are known to interact *in vitro* (*Kleaveland et al., 2009*), we next tested whether Heg1 protein affects protein levels of its binding partner Krit1. To this end, we first generated a transgenic line of zebrafish for expression of EGFP-Krit1 [*Tg(UAS:EGFP-krit1)$^{pbb21}$*]. Next, we analyzed the levels of EGFP-Krit1 protein in *heg1* MO-injected embryos and found that they were significantly lower than in WT control embryos in the heart (*Figure 1C–E*) and the caudal vasculature at 30 hpf (*Figure 1—figure supplement 2*). Taken together, the levels of *heg1* mRNA expression are positively regulated by blood-flow and Klf2a-dependent mechanotransduction. In turn, Heg1 has a stabilizing effect on Krit1 levels.

## Overexpression of *heg1* or *krit1* mRNA dampens expression levels of *klf2a* mRNA

A loss of Ccm proteins in zebrafish or mice results in higher levels of *klf2* mRNA expression, suggesting that the physiological role of Ccm proteins is to modulate *klf2* expression levels in response to blood-flow (*Renz et al., 2015*; *Zhou et al., 2015*, *2016*). Given that levels of *heg1* mRNA expression are affected by blood-flow, we explored whether upregulation of Heg1 or Krit1 would have an impact on endothelial mechanotransduction pathways. First, we injected *heg1* mRNA at the one-cell stage and assessed *klf2a* mRNA levels at 24 and 48 hpf. At both times, high levels of *heg1* mRNA correlated with significantly lower levels of *klf2a* expression (*Figure 2A*). Next, we tested whether overexpression of *krit1* also downregulates levels of *klf2a* mRNA by using a transgenic line with heat-shock-inducible *krit1* [*Tg(hsp70l:krit1_IRES EGFP)$^{md6}$*]. Treatment with multiple heat-shocks at

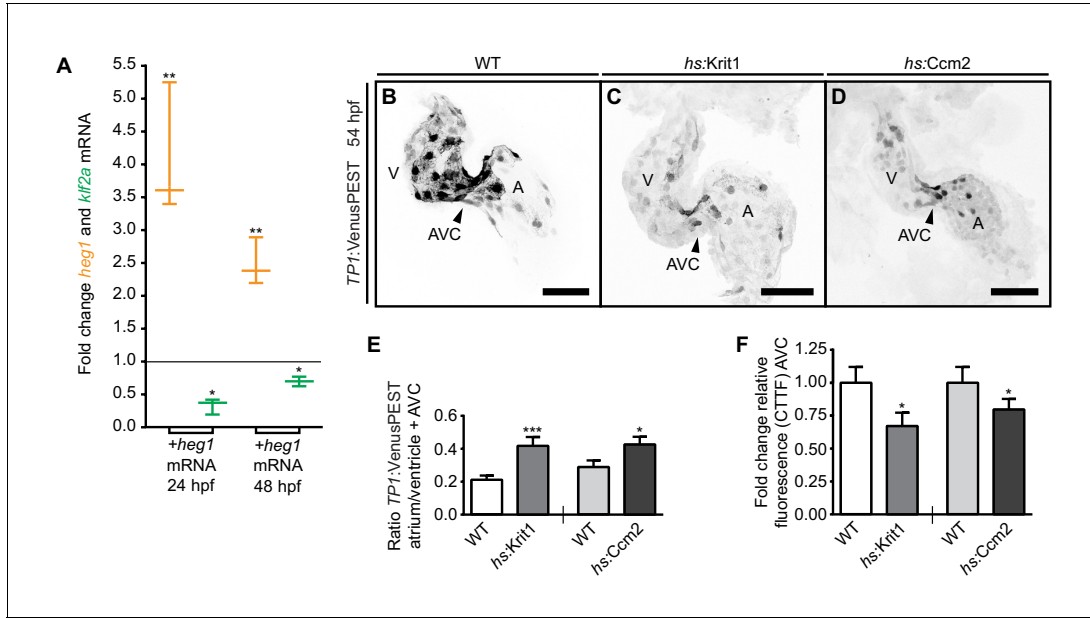

**Figure 2.** Desensitization of endocardial cells to blood-flow by *heg1*, *krit1*, or *ccm2* overexpression. (A) Correlation of high *heg1* and low *klf2a* mRNA expression levels at 24 and 48 hpf as determined by RT-qPCR. (B–D) Projections of confocal z-stack images of 54 hpf hearts expressing the flow-responsive marker *Tg(TP1:VenusPEST)$^{s940}$* in heat-shocked wild-type (WT) (B), heat-shock- (hs) induced *krit1* overexpression (C), and hs-induced *ccm2* overexpression (D). (E) Different ratios of corrected total tissue fluorescence (CTTF) of atrial versus ventricular (+atrioventricular canal region (AVC), arrowhead) expression of the flow-responsive marker *Tg(TP1:VenusPEST)$^{s940}$* between 54 hpf WT and *krit1*-overexpressing hearts (n = 37) and WT and *ccm2*-overexpressing hearts (n = 39). *krit1*- or *ccm2*-overexpression results in more equal chamber expression of flow-responsive marker *Tg(TP1:VenusPEST)$^{s940}$* expression (depicted is a WT heart corresponding to the *krit1*-overexpression experiment). (F) Fold change of relative fluorescence (CTTF) in the AVC of *Tg(TP1:VenusPEST)$^{s940}$* in *krit1*- and *ccm2*- overexpressing hearts versus the corresponding controls. Scale bars are 50 μm. Mean values ± SEM are shown. Unpaired t-test was used to compare each condition with the WT (*p<0.05, **p<0.01, ***p<0.001).
DOI: https://doi.org/10.7554/eLife.28939.005

The following figure supplement is available for figure 2:

**Figure supplement 1.** Exclusive endocardial expression of the Notch reporter *Tg(TP1:VenusPEST)$^{s940}$* and cardiac cushions forming normally on overexpression of *krit1* and *ccm2*.
DOI: https://doi.org/10.7554/eLife.28939.006

14, 24, and 40 hpf resulted in significant downregulation of *klf2a* mRNA levels throughout the entire embryo at 48 hpf as determined by RT-qPCR (*klf2a*: fold change 0.63, p<0.05, n = 4 replicates; *krit1*: fold change 9.77, p<0.001, n = 4 replicates). Cardiac cushions formed normally after this treatment (*Figure 2—figure supplement 1*) and blood-flow was not affected as assessed by visual inspection (n > 100 embryos analyzed showed normal blood circulation). Hence, the forced upregulation of *heg1* or *krit1* dampens expression levels of *klf2a* mRNA. This finding suggests that the strength of the biomechanical forces resulting from blood-flow impacts the expression levels of Ccm proteins; this modulates mechanosensitive signaling within endothelial cells upstream of *klf2a*.

## Krit1 and Ccm2 impact expression of the mechanosensitive Notch activity reporter *Tg(TP1:VenusPEST)*

To further assess the physiological effects of altered levels of CCM proteins for endocardial patterning, we analyzed expression levels of the transgenic line *Tg(TP1:VenusPEST)*[s940], which expresses destabilized Venus protein in regions of strong Notch activity (*Ninov et al., 2012*). In zebrafish, *notch1b* is a blood-flow-responsive gene that is highly expressed at cardiac cushions (*Vermot et al., 2009*). By 54 hpf, the expression of *Tg(TP1:VenusPEST)*[s940] is restricted to the ventricular chamber and is particularly strong within the AVC region (*Figure 2B*). To induce upregulation of *krit1* or *ccm2*, *Tg(hsp70l:krit1_IRES EGFP)*[md6] or *Tg(hsp70l:ccm2_IRES EGFP)*[md12] embryos were treated with multiple heat-shocks at 14, 24, 40, and 48 hpf. On *krit1* or *ccm2* overexpression, and consistent with a dampening of blood-flow responses, expression from the *Tg(TP1:VenusPEST)*[s940] reporter was generally weakened and was not restricted to the ventricle and AVC region at 54 hpf (*Figure 2B–F*; *Figure 2—figure supplement 1*). Hence, overexpression of *krit1* or *ccm2* affects the expression levels and pattern of the *Tg(TP1:VenusPEST)*[s940] reporter, which is responsive to blood-flow within the heart. Notably, the expression of this Notch reporter was exclusively endocardial in all of these experimental conditions (*Figure 2—figure supplement 1*). This result suggests that endocardial cells become desensitized to blood-flow-induced mechanosensitive signaling when *krit1* or *ccm2* are overexpressed.

## Krit1 is required for generation of abluminal cell fates during valvulogenesis

Given the strong involvement of Ccm proteins in mechanotransduction pathways, we next explored whether they have a developmental role during remodeling of cardiac cushions into valve leaflets, a morphogenetic process that is highly sensitive to the biomechanical stimulus of blood-flow (*Beis et al., 2005*; *Pestel et al., 2016*; *Scherz et al., 2008*; *Steed et al., 2016*; *Vermot et al., 2009*). Previously, it was not possible to address such a potential developmental role as zebrafish *ccm* mutant endocardial cells fail to form cardiac cushions [*Figure 3B*; (*Renz et al., 2015*). This is in contrast to WT endocardial cushion cells that acquire cuboidal shapes at 48 hpf and express activated leukocyte cell adhesion molecule (Alcam) (*Figure 3A*; *Beis et al., 2005*). To elucidate whether CCM proteins play a role in this process, we performed a rescue experiment by injecting mRNA encoding EGFP-Krit1 into *krit1*[ty219c] mutants, which rescued the cardiovascular defects associated with the loss of Krit1 at 48 hpf (*Figure 3C*, *Figure 3—figure supplement 1*). Zebrafish *ccm* mutants have increased endocardial cell numbers by 48 hpf (*Renz et al., 2015*). To determine whether injection of mRNA encoding EGFP-Krit1 into *krit1*[ty219c] mutants could rescue endocardial cell numbers, we compared ventricular endocardial cell numbers of WT and *krit1*[ty219c] rescued embryos at 48 and 55 hpf (*Figure 3—figure supplement 2*). Consistent with the strong rescue of the *krit1*[ty219c] mutant phenotype at 48 hpf, the number of ventricular endocardial cells was normal at this stage. However, by 55 hpf, ventricular endocardial cell numbers had increased beyond those in WT, which suggests that at this stage the *krit1*[ty219c] mutant phenotype becomes expressed again. To assay whether the re-appearance of a late *krit1*[ty219c] mutant phenotype correlated with decreasing *egfp-krit1* mRNA levels, we monitored *krit1* mRNA levels over time using RT-qPCR. Indeed, WT embryos injected with *egfp-krit1* mRNA showed a significant decrease of *krit1* mRNA levels between 24 and 96 hpf (*Figure 3—figure supplement 3*). This result is consistent with recurrence of the *krit1*[ty219c] mutant phenotype at 55 hpf.

During the stage at which cardiac cushions form, we selected *egfp-krit1* mRNA injected *krit1*[ty219c] mutant embryos with normal blood-flow and found that they had developed normal endocardial

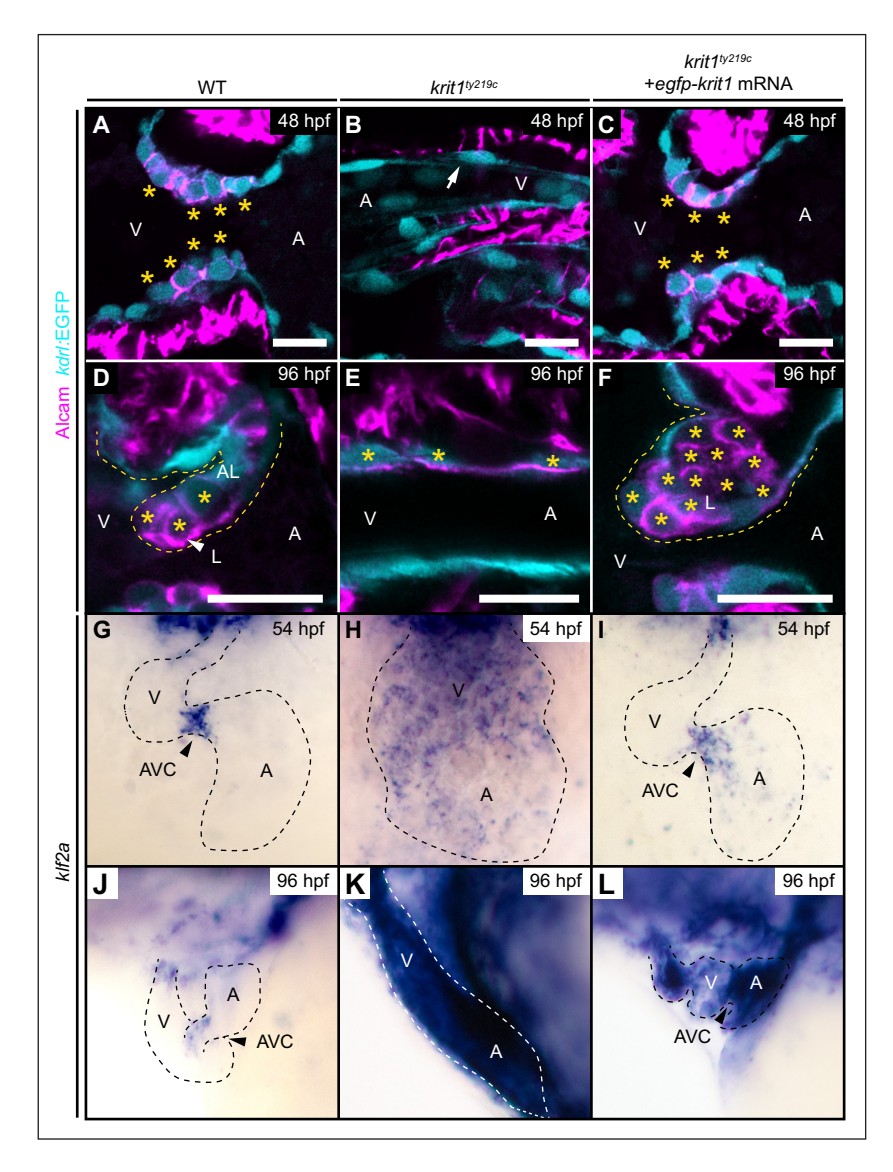

**Figure 3.** Krit1 is required for abluminal cell fates during valvulogenesis. (A–F) Single confocal z-section images of endocardial cells within the atrioventricular canal (AVC) region marked by $Tg(kdrl:EGFP)^{s843}$ expression (cyan) and Alcam staining (magenta). (A) At 48 hpf, wild-type (WT) endocardial cushion cells express Alcam (asterisks). (B) In $krit1^{ty219c}$ mutants, endocardial cells of the AVC region do not express Alcam and cushions do not form (arrow). (C) Injection of $egfp$-$krit1$ mRNA rescues cardiac cushion formation in $krit1^{ty219c}$ mutants (n = 15/15). $krit1^{ty219c}$ mutant hearts form cardiac cushions and endocardial cushion cells express Alcam. (D) By 96 hpf, Alcam expression is mainly restricted to luminal endocardial cells of the developing WT cardiac valve leaflet (asterisks). (E) $krit1^{ty219c}$ mutants do not form valve leaflets but AVC endocardial cells express Alcam (asterisks). (F) $krit1^{ty219c}$ mutant that was initially rescued by $egfp$-$krit1$ mRNA injection has a dysmorphic cardiac valve leaflet at 96 hpf with an agglomeration of luminal Alcam-positive cells (asterisks). (G–L) Whole-mount in situ hybridization of $klf2a$ cardiac expression. (G) At 54 hpf, $klf2a$ expression is restricted to endocardial cells of the AVC in WT while (H) $klf2a$ is strongly expressed throughout the entire endocardium in $krit1^{ty219c}$ mutant. (I) $krit1^{ty219c}$ mutant rescued by injection of $egfp$-$krit1$ mRNA has a restricted and strong $klf2a$ expression at the AVC (n = 6/6). (J) By 96 hpf, $klf2a$ expression is restricted to the AVC in WT while (K) $klf2a$ is strongly expressed throughout the entire endocardium in $krit1^{ty219c}$ mutants. (L) $krit1^{ty219c}$ mutant embryo injected with $egfp$-$krit1$ mRNA has high $klf2a$ levels throughout the entire endocardium (n = 8/8). A: atrium, V: ventricle, L: luminal, AL: abluminal. Scale bars are 10 μm.
DOI: https://doi.org/10.7554/eLife.28939.007

The following figure supplements are available for figure 3:

*Figure 3 continued on next page*

*Figure 3 continued*

**Figure supplement 1.** Rescue of *krit1^{ty219c}* mutant cardiovascular phenotypes by injection of mRNA encoding EGPF-Krit1.

DOI: https://doi.org/10.7554/eLife.28939.008

**Figure supplement 2.** Injection of mRNA encoding EGFP-Krit1 into *krit1^{ty219c}* mutants prevents overgrowth of ventricular endocardial cells.

DOI: https://doi.org/10.7554/eLife.28939.009

**Figure supplement 3.** *krit1* expression decreases to a basal level by 96 hpf in rescued embryos.

DOI: https://doi.org/10.7554/eLife.28939.010

**Figure supplement 4.** Cadherin-5 is misexpressed in *krit1^{ty219c}* mutant embryonic hearts.

DOI: https://doi.org/10.7554/eLife.28939.011

**Figure supplement 5.** Krit1 is required for abluminal cell fates during cardiac valve leafet formation.

DOI: https://doi.org/10.7554/eLife.28939.012

cushions by 48 hpf (*Figure 3C*). Next, we analyzed whether these rescued *krit1^{ty219c}* mutant embryos would revert to a mutant cardiovascular phenotype by 96 hpf, a point at which endocardial cushions have been transformed via complex cellular rearrangements into double-layered valve leaflets (*Beis et al., 2005*; *Pestel et al., 2016*; *Scherz et al., 2008*; *Steed et al., 2016*). In WT, those endocardial cells that face the luminal side of the valve leaflet and are exposed to the strongest shear stress, express high levels of *klf2a* (*Steed et al., 2016*) and can be identified by strong Alcam (*Figure 3D*) and Cdh5 expression (*Steed et al., 2016*) (*Figure 3—figure supplement 4A'–A'''*). In comparison, endocardial cells on the abluminal side of the valve leaflet have lower levels of *klf2a*, (*Steed et al., 2016*) and Alcam (*Figure 3D*) and Cdh5 are only weakly expressed in abluminal cells (*Figure 3—figure supplement 4A'–A'''*). This is in tune with the more mesenchymal identity ascribed to this endocardial cell population (*Pestel et al., 2016*; *Steed et al., 2016*). We found that *krit1^{ty219c}* mutant embryos with a cardiac rescue that were exposed to normal blood-flow at 48 hpf, developed strongly dysmorphic valve leaflets by 96 hpf (n = 17/23 *krit1^{ty219c}* mutant embryos). At that stage, most mutants had valve leaflet endocardial cells on the abluminal sides that strongly expressed high levels of Alcam similar to luminal *klf2a*-positive cells in WT (*Figure 3F*; n = 17/23 *krit1^{ty219c}* mutant embryos). In addition, Cdh5 colocalized with Alcam on the abluminal side of dysmorphic valve leaflets (*Figure 3—figure supplement 4C'–C'''*). This phenotype corresponded with an agglomeration of endocardial cushion cells with characteristics of luminal cells. We ruled out that the observed phenotype resulted from an *egfp-krit1* overexpression effect because *krit1* mRNA levels in these rescued embryos had decreased to a basal level of expression by 96 hpf (*Figure 3—figure supplement 3*). In tune with this observation, untreated *krit1^{ty219c}* mutants express high levels of Cdh5 throughout the entire endocardium at 96 hpf (*Figure 3—figure supplement 4B–B'''*) and Alcam is expressed at high levels within the AVC region (*Figure 3E*, *Figure 3—figure supplement 4B'–B'''*). Only few *krit1^{ty219c}* mutant embryos had a normal distribution of Alcam-positive luminal endocardial cells by 96 hpf, a phenotype more similar to WT (n = 6/23 *krit1^{ty219c}* mutant embryos; *Figure 3—figure supplement 5*). Taken together, the loss of Krit1 mostly resulted in a failure of endocardial cells to acquire mesenchymal-like fates and to generate an abluminal population of valve leaflet cells.

## Krit1 dampens *klf2a* expression and Notch activation during cardiac valve morphogenesis

Ingression of abluminal cells from cardiac cushions has been associated with a bias in *klf2a* expression levels: whereas some endocardial cushion cells express high levels of *klf2a* and contribute to the luminal part of the valve leaflet, others express lower levels of *klf2a* which correlates with a mesenchymal-like morphology and their contribution to the abluminal portion of the valve leaflet (*Steed et al., 2016*). Our findings suggest that elevated levels of *klf2a* mRNA are the cause for defects in valve leaflet morphogenesis. To assess *klf2a* expression in rescued *krit1^{ty219c}* mutant hearts, we used whole-mount *in situ* hybridization. By 54 hpf, *klf2a* was strongly expressed within cardiac cushions at the AVC, whereas in other regions of the endocardium expression levels were lower or undetectable (*Figure 3G*). In *krit1^{ty219c}* mutants, *klf2a* was strongly expressed throughout the entire endocardium and was not restricted to the AVC (*Figure 3H*). At 54 hpf, in *krit1^{ty219c}*

mutant embryos that were rescued by injection of mRNA encoding EGFP-Krit1, the expression of *klf2a* had a normal pattern with a localized strong expression at the AVC cushions (n = 6/6) (*Figure 3I*). However, by 96 hpf, the expression of *klf2a* was not restricted to the AVC and outflow tract regions of the endocardium as in WT (*Figure 3J*) but had reverted to wide and strong expressionpresent throughout the entire endocardium (n = 8/8) (*Figure 3L*).

*klf2a* is a regulator of *notch1b* expression at cardiac cushions (*Vermot et al., 2009*). In addition, several studies have demonstrated that the reduction of CCM protein levels is associated with decreased Notch activity in human endothelial cells (*Wüstehube et al., 2010*; *You et al., 2013*). Hence, changes in Notch activity may contribute to defective cardiac valve leaflet morphogenesis in *ccm* mutants. To address this question, we first assessed *notch1b* expression by whole-mount *in situ* hybridization. At 48 hpf, *notch1b* was strongly expressed within high shear stress regions including the AVC of the WT heart (*Figure 4A*). In contrast, *notch1b* was expressed throughout the entire endocardium in *krit1^{ty219c}* mutant embryos (*Figure 4B*). Next, we characterized Notch signaling activity using the *Tg(TP1:VenusPEST)^{s940}* reporter. In WT, Notch activity was in a mosaic pattern within endocardial cushions and a few cushion cells neighboring the ventricle were consistently downregulating Notch activity (*Figure 4C'*; n > 30 embryos analyzed). Those cells lacking Notch activity were in positions that have been shown to initiate endocardial sprouting behaviors and to contribute to formation of the abluminal portion of the cardiac valve leaflets (*Steed et al., 2016*). In striking contrast, in *ccm2^{m201}* and *krit1^{ty219c}* mutants, Notch reporter expression was expanded throughout large regions of the endocardium (*Figure 4D*, *Figure 4—figure supplement*

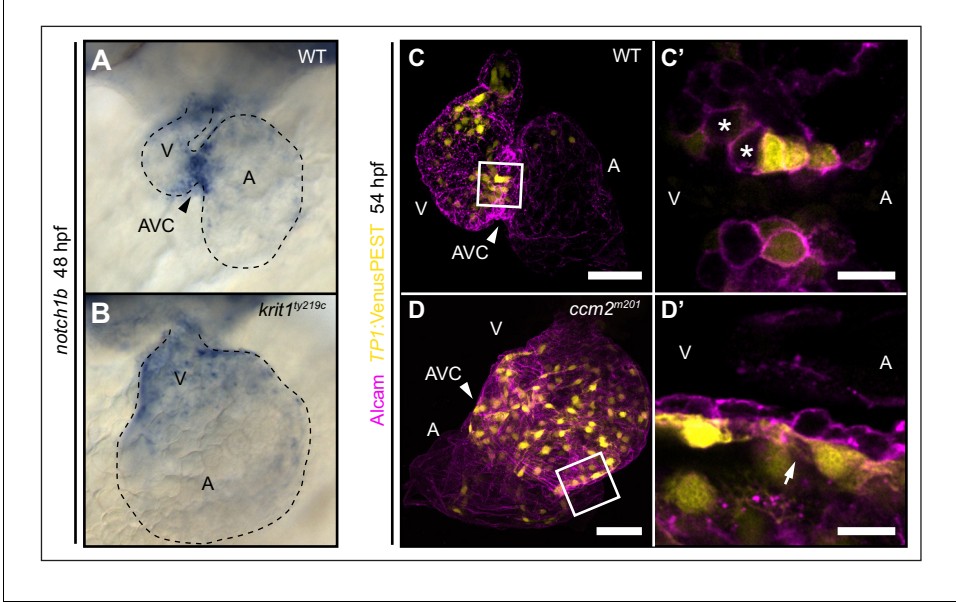

**Figure 4.** Notch activity within endocardial cushion cells is affected in *ccm* mutants. (A, B) Whole-mount *in situ* hybridization of *notch1b* cardiac expression at 48 hpf. (A) *notch1b* expression is restricted to endocardial cells of the atrioventricular canal (AVC) in wild-type (WT), while *notch1b* is strongly expressed throughout all ventricular cells in *krit1^{ty219c}* mutants (B). (C, D) Projections of single confocal z-section images of endocardial cells marked by *Tg(TP1:VenusPEST)^{s940}* expression (yellow) and Alcam staining (magenta) at 54 hpf. (C) In WT, Notch activity is highest in endocardial cells of the AVC and outflow regions. (C') Single confocal plane section of the AVC (white box in C) reveals that some endocardial cells close to the ventricle lack Notch activity (asterisks). (D) In *ccm2^{m201}* mutants, the domain of high Notch activity is expanded to most ventricular endocardial cells. (D') Single confocal plane section of the AVC region (box in D, arrow) shows high Notch expression in all endocardial cells of the AVC region. A: atrium, V: ventricle, Scale bars are 50 μm (C, D), and 10 μm (C', D').
DOI: https://doi.org/10.7554/eLife.28939.013
The following figure supplement is available for figure 4:

**Figure supplement 1.** The region of Notch activity within the endocardium is expanded in *krit1^{ty219c}* mutants.
DOI: https://doi.org/10.7554/eLife.28939.014

*1B*), and was especially active within all endocardial cells at the AVC region which demonstrated that a singling-out process and downregulation of Notch signaling among few cushions cells did not occur (*Figure 4D'*; n > 30 embryos analyzed). In comparison, the expression of *tbx2b*, another AVC marker gene (*Sedletcaia and Evans, 2011*), was not expanded in *ccm2^m201* mutants (*Figure 4—figure supplement 1C–D*). Hence, expansion of *notch1b* expression domain in *ccm2^m201* or *krit1^ty219c* mutants is not a result of regional expansion of the AVC. Taken together, our findings suggest a critical role for Krit1 in modulating expression levels of *klf2a* and for Notch signaling among endocardial cushion cells which may control endocardial sprouting during the formation of cardiac valve leaflets.

## Discussion

Our study establishes CCM proteins as having an important role in endothelial mechanosensitive signaling. We report that levels of *heg1* mRNA expression are positively regulated by blood-flow in a Klf2-dependent manner and that Heg1 stabilizes Krit1 protein levels. We also show that overexpression of Heg1 and Krit1 dampens expression of *klf2a* mRNA. This establishes a negative feedback loop that desensitizes endothelial cells to blood-flow-induced mechanosensitive signaling by dampening expression levels of *klf2a*. The fine-tuning of blood-flow-sensitive responses is particularly important during developmental processes such as cardiac valvulogenesis, which involves establishment of regionalized differences in *klf2a* expression levels (*Steed et al., 2016*). While the endocardial cells facing the lumen of the heart experience high fluid shear forces, and consequently begin to express high levels of *klf2a*, those on the other side do not. In tune with this observation, the process of valvulogenesis is critically dependent on the biophysical stimulus by blood-flow (*Beis et al., 2005*; *Heckel et al., 2015*; *Pestel et al., 2016*; *Steed et al., 2016*) and is affected by the loss of *klf2a* (*Steed et al., 2016*). Hence, mechanosensitive signaling pathways actively reshape cardiac valve leaflets in a manner that is highly sensitive to constantly changing blood-flow conditions. The present work suggests that Krit1 plays an important role in modulating expression levels of *klf2a* during this complex morphogenetic process. Our results suggest that downregulation of endocardial mechanosensitivity by CCM proteins generates a bias in expression levels of *klf2a* which is critical for correct valvulogenesis.

In accordance with our findings, expression of the KRIT1-CCM2-associated protein CCM2L has also been reported in a blood-flow-dependent manner (*Cullere et al., 2015*). In a complex with KRIT1 and CCM2, CCM2L directly binds to the MAP kinase MEKK3 and inhibits its activation. This reduces the ability of MEKK3 to phosphorylate MEK5 which is required to activate ERK5 (*Cullere et al., 2015*). The MEKK3-MEK5-ERK5 pathway is involved in activation of *KLF2* expression (*Zhou et al., 2015*; *Zhou et al., 2016*), the control of endoMT (reviewed in *Drew et al., 2012*), and murine cardiovascular development (*Yang et al., 2000*; reviewed in *Rose et al., 2010*). Activation of *klf2a* expression and control of endoMT processes could be particularly important during formation of functional cardiac valve leaflets. Within abluminal endocardial cells of the zebrafish cardiac valve leaflet, the loss of Krit1 prevents downregulation of Cdh5 and Alcam proteins, a function that might ultimately be related to elevated levels of Klf2a. As yet, no blood-flow-independent mechanism has been identified that would trigger such an upregulation of Klf2a in the absence of Ccm proteins (see model, *Figure 5*). As Klf2a regulates the expression of endocardial *notch1b* within endocardial cushions (*Vermot et al., 2009*), the formation of abluminal valve leaflets may be regulated by a Notch-dependent lateral inhibition process similar to angiogenic sprouting of tip cells. Whereas high *klf2a* expression corresponds to expression of the endothelial junctional proteins Cdh5 and Alcam within luminal endocardial valve cells, abluminal endocardial cells that are shielded from blood-flow and express lower levels of *klf2a* and *notch1b* downregulate the expression of Cdh5 (*Steed et al., 2016*) and Alcam and acquire a more mesenchymal-like state. Several publications have discussed the possibility that downregulation of Cdh5 and Alcam in abluminal endocardial cells and the behaviour of these endocardial cells during zebrafish valvulogenesis (*Beis et al., 2005*; *Lagendijk et al., 2011*; *Scherz et al., 2008*; *Steed et al., 2016*) is highly reminiscent of the endoMT process that occurs during cardiac valve formation in the mouse (*Chiplunkar et al., 2013*). Our findings now implicate CCM proteins in providing a bias in *klf2a* and *notch1b* expression levels within cardiac cushions, which may be further elaborated by Notch-dependent lateral inhibition that could allocate cells to an abluminal fate (see model, *Figure 5*). In tune with such a model, Notch activity was in a mosaic pattern

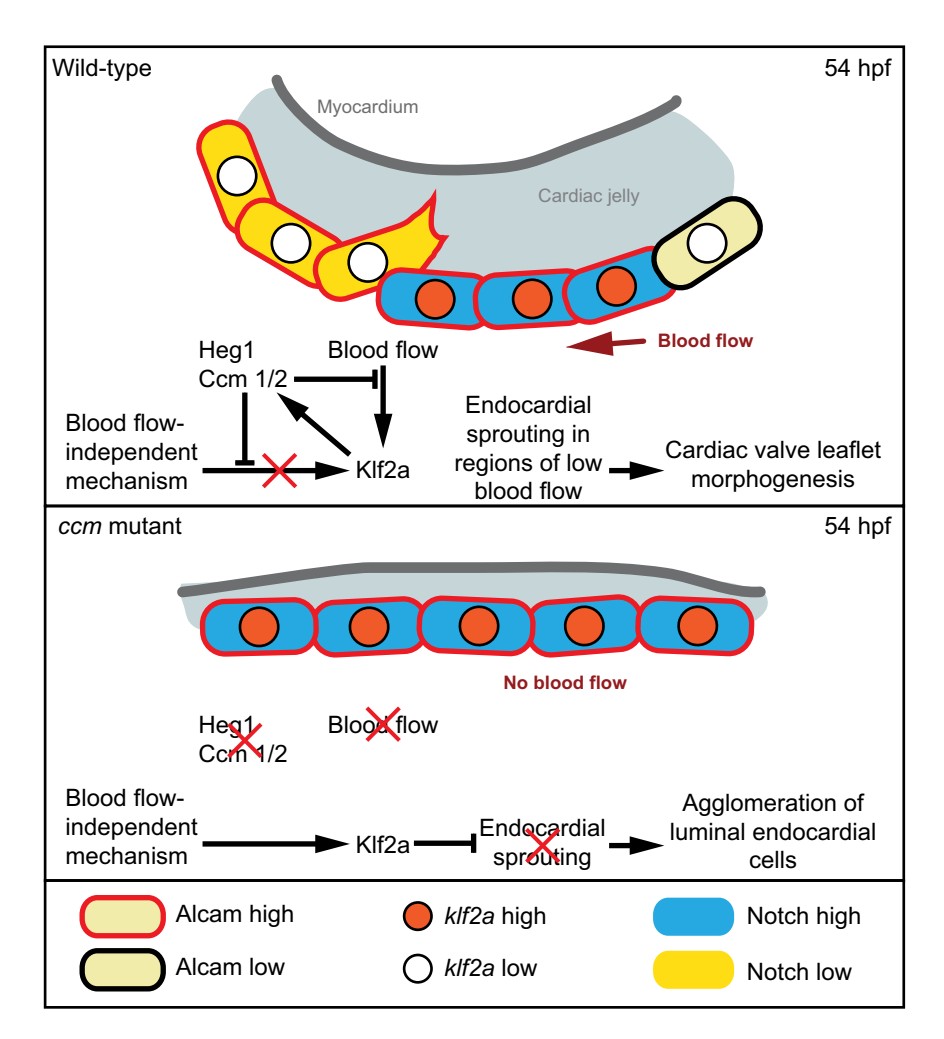

**Figure 5.** Model of the molecular pathways involved in endocardial cushion cell sprouting at 54 hpf. In wild-type, endocardial cells exposed to blood-flow have high levels of Klf2a, Notch, Alcam, and maintain cell adhesion. Ccm proteins provide a bias in Klf2a and Notch expression levels. Some cells that express lower levels of *klf2a* and have lower Notch activity establish protrusions and migrate into the cardiac jelly (endocardial sprouting). In *ccm* mutants, endocardial cells overexpress *klf2a* and have higher Notch activity because of a blood-flow-independent mechanism. As a consequence, high levels of Notch activity throughout the entire endocardium result in failure of endocardial sprouting.

DOI: https://doi.org/10.7554/eLife.28939.015

within WT endocardial cushions, while in *ccm* mutants, Notch signaling was widely active. Interestingly, pharmacological inhibition of Notch signalling has been shown to cause similar cardiac valve leaflet morphogenesis defects (***Beis et al., 2005***). Hence, it remains an important question for future research to elucidate whether Notch-dependent endocardial sprouting morphogenesis and/or endoMT are involved in the process of valve leaflet formation.

Taken together, our findings uncover crucial roles of Heg1 and Krit1 in controlling the sensitivity of endothelial cells to hemodynamic forces. This may promote developmental adaptations in response to changes in the strength of hemodynamic forces that occur during cardiovascular development. Within the early zebrafish heart tube, shear forces resulting from blood-flow increase rapidly within the first 4–5 days post fertilization (***Hove et al., 2003***). Similar changes in blood-flow dynamics also occur during the remodelling of the vasculature or during pathological changes of

blood-flow patterns. It will be important to assess the physiological roles of CCM proteins during these different adaptations of the vasculature.

# Materials and methods

**Key resources table**

| Reagent type (species) or resource | Designation | Source or reference | Identifiers | Additional information |
|---|---|---|---|---|
| gene (heg1) | heg1 | ZFIN | ZFIN ID: ZDB-GENE-040714–1 | |
| gene (krit1) | krit1 | ZFIN | ZFIN ID: ZDB-GENE-030131–555 | |
| gene (klf2a) | klf2a | ZFIN | ZFIN ID: ZDB-GENE-011109–1 | |
| gene (klf2b) | klf2b | ZFIN | ZFIN ID: ZDB-GENE-011109–2 | |
| gene (tnnt2a) | tnnt2a | ZFIN | ZFIN ID: ZDB-GENE-000626–1 | |
| gene (ccm2) | ccm2 | ZFIN | ZFIN ID: ZDB-GENE-040712–6 | |
| gene (notch1b) | notch1b | ZFIN | ZFIN ID: ZDB-GENE-990415–183 | |
| strain, strain background (krit1ty219c) | krit1ty219c | DOI: 10.1242/dev.02469 | ZFIN ID: ZDB-ALT-980203–1289 | |
| strain, strain background (ccm2m201) | ccm2m201 | PMID:9007227 | ZFIN ID: ZDB-ALT-980203–523 | |
| strain, strain background (Tg(EPV.Tp1-Mmu.Hbb: Venus-Mmu.Odc1)s940) | Tg(EPV.Tp1-Mmu.Hbb: Venus-Mmu.Odc1)s940 | DOI: 10.1242/dev.076000 | ZFIN ID: ZDB-ALT-120419–6 | |
| strain, strain background (Tg(kdrl:EGFP)s843) | Tg(kdrl:EGFP)s843 | DOI: 10.1242/dev.02087 | ZFIN ID: ZDB-ALT-050916–14 | |
| strain, strain background (Tg(fli1a:GAL4FF)ubs3) | Tg(fli1a:GAL4FF)ubs3 | DOI: 10.1016/j.cub. 2011.10.016 | ZFIN ID: ZDB-ALT-120113–6 | |
| strain, strain background (Tg(hsp70l:Krit1_IRES_EGFP)md6) | Tg(hsp70l: Krit1_IRES_EGFP)md6 | this paper | | |
| strain, strain background (Tg(hsp70l:Ccm2_IRES_EGFP)md12) | Tg(hsp70l: Ccm2_IRES_EGFP)md12 | this paper | | |
| strain, strain background (Tg(hsp70l:Klf2a_IRES_EGFP)pbb22) | Tg(hsp70l: Klf2a_IRES_EGFP)pbb22 | this paper | | |
| strain, strain background (Tg(UAS:EGFP-Krit1, cryaa:EGFP)pbb21) | Tg(UAS:EGFP-Krit1, cryaa:EGFP)pbb21 | this paper | | |
| genetic reagent (tnnt2a morpholino) | tnnt2a MO | DOI: 10.1038/ng875 | ZFIN ID: ZDB-MRPHLNO-060317–4 | |
| genetic reagent (klf2a morpholino) | klf2a MO | DOI: 10.1038/nature08889 | ZFIN ID: ZDB-MRPHLNO-100610–9 | 2 ng/embryo |
| genetic reagent (klf2b morpholino) | klf2b MO | DOI: 10.1016/j.devcel. 2014.12.016 | ZFIN ID: ZDB-MRPHLNO-150427–1 | 1 ng/embryo |
| genetic reagent (heg1 morpholino) | heg1 MO | PMID:14680629 | ZFIN ID: ZDB-MRPHLNO-080714–5 | 5 ng/embryo |
| genetic reagent (klf2a probe) | klf2a | DOI: 10.1016/j.devcel. 2014.12.016 | ZFIN ID: ZDB-FIG-150407–1 | 5 ng/embryo |
| genetic reagent (notch1b probe) | notch1b | DOI: 10.1126/science. 293.5535.1670 | ZFIN ID: ZDB-FIG-151113–25 | |
| genetic reagent (heg1 probe) | heg1 | DOI: 10.1242/dev.143362 | ZFIN ID: ZDB-PUB-031217–1 | |
| genetic reagent (tbx2 probe) | tbx2 | DOI: 10.1002/dvdy.22622 | ZFIN ID: ZDB-PUB-110502–3 | |
| Antibody (rabbit anti-VE-Cadherin (Cdh5)) | Cdh5 | DOI: 10.1016/j.ydbio. 2008.01.038 | ZFIN ID: ZDB-PUB-080326–18 | 1:200 |

*Continued on next page*

*Continued*

| Reagent type (species) or resource | Designation | Source or reference | Identifiers | Additional information |
|---|---|---|---|---|
| Antibody (mouse anti-Zn-8/Alcam) | Alcam | Developmental Studies Hybridoma Bank | ZFIN ID: ZDB-ATB-081002–22 | 1:25 |
| Antibody (mouse anti-Myh6) | Myh6 | Developmental Studies Hybridoma Bank | ZFIN ID: ZDB-ATB-081002–54 | 1:10 |
| Antibody (Alexa Fluor 633-conjugated goat anti-rabbit) | Alexa Fluor 633-conjugated goat anti-rabbit | Invitrogen A21070 | | 1:200 |
| Antibody (Rhodamine Red- X-conjugated goat anti-mouse) | Rhodamine Red- X-conjugated goat anti-mouse | Jackson ImmunoResearch Laboratories 115-295-003 | | 1:200 |
| Antibody (Dylight 649-conjugated goat anti-mouse) | Dylight 649-conjugated goat anti-mouse | Jackson ImmunoResearch Laboratories 115-495-003 | | 1:200 |
| recombinant DNA reagent (pDestTol2 (#426)) | pDestTol2 | N. Lawson | Lawson Lab: #426 | |
| recombinant DNA reagent (p5E-hsp70l (#222)) | p5E-hsp70l | N. Lawson | Lawson Lab: #222 | |
| recombinant DNA reagent (p5E-CMV/SP6 (#382)) | p5E-CMV/SP6 | Chien, Univ. Utah | Tol2kit: #382 | |
| recombinant DNA reagent (p5E-UAS (#327)) | p5E-UAS | Chien, Univ. Utah | Tol2kit: #327 | |
| recombinant DNA reagent (p3E-IRES_EGFPpA (#389)) | p3E-IRES_EGFPpA | N. Lawson | Lawson Lab: #389 | |
| recombinant DNA reagent (p3E-cryaa:EGFPpA) | p3E-cryaa:EGFPpA | other | | plasmid was generated in our lab |
| recombinant DNA reagent (p3E-EGFPpA (#366)) | p3E-EGFPpA | Chien, Univ. Utah | Tol2kit: #366 | |
| recombinant DNA reagent (p3E-pA (#383)) | p3E-pA | Chien, Univ. Utah | Tol2kit: #383 | |
| recombinant DNA reagent (pDest_Tol2pA_Hsp70l: Krit1_IRES_EGFPpA) | pDest_Tol2pA_Hsp70l: Krit1_IRES_EGFPpA | this paper | | |
| recombinant DNA reagent (pDest_Tol2pA_Hsp70l: Ccm2_IRES_EGFPpA) | pDest_Tol2pA_Hsp70l: Ccm2_IRES_EGFPpA | this paper | | |
| recombinant DNA reagent (p3E-pA (pDest_Tol2pA_ Hsp70l:Klf2a_IRES_EGFPpA) | pDest_Tol2pA_Hsp70l: Klf2a_IRES_EGFPpA | this paper | | |
| recombinant DNA reagent (pDest_Tol2pA_UAS: EGFP-Krit1pA,cryaa:EGFPpA) | pDest_Tol2pA_UAS: EGFP-Krit1pA,cryaa:EGFPpA | this paper | | |
| recombinant DNA reagent (pDest_Tol2pA_CMV/SP6: EGFP-Krit1pA) | pDest_Tol2pA_CMV/SP6: EGFP-Krit1pA | this paper | | |
| sequence-based reagent (qPCR-primer heg1 FW) | *heg1* FW | this paper | | |
| sequence-based reagent (qPCR-primer heg1 RW) | *heg1* RW | this paper | | |
| sequence-based reagent (qPCR-primer krit1 FW) | *krit1* FW | this paper | | |
| sequence-based reagent (qPCR-primer krit1 RW) | *krit1* RW | this paper | | |
| sequence-based reagent (qPCR-primer klf2a FW) | *klf2a* FW | this paper | | |

*Continued on next page*

*Continued*

| Reagent type (species) or resource | Designation | Source or reference | Identifiers | Additional information |
|---|---|---|---|---|
| sequence-based reagent (qPCR-primer klf2a RW) | *klf2a* RW | this paper | | |
| sequence-based reagent (qPCR-primer eif1b FW) | *eif1b* FW | this paper | | |
| sequence-based reagent (qPCR-primer eif1b RW) | *eif1b* RW | this paper | | |
| commercial assay or kit (SP6 polymerase (mMessage Machine kit, Ambion)) | SP6 polymerase | Ambion | Ambion:AM1340 | |
| commercial assay or kit (RevertAid H Minus First Strand cDNA Synthesis kit (ThermoFisher Scientific)) | RevertAid H Minus First Strand cDNA Synthesis kit | ThermoFisher Scientific | ThermoFisher Scientific:K1631 | |
| commercial assay or kit (KAPA Sybr Fast qPCR kit (Peqlab)) | KAPA Sybr Fast qPCR kit | Peglab | Peglab:4385612 | |
| chemical compound, drug (1- phenyl-2-thiourea (PTU)) | PTU | Sigma Aldrich | Sigma Aldrich:P7629 | 0.003% |
| chemical compound, drug (Rhodamine-Phalloidin) | Rhodamine-Phalloidin | Invitrogen | Invitrogen:R415 | 1:250 |
| chemical compound, drug (Tricaine (3-amino benzoic acidethylester)) | Tricaine | Sigma-Aldrich | Sigma Aldrich:A-5040 | 0.16 mg/ml |
| software, algorithm (GraphPad Prism6) | GraphPad Prism6 | GraphPad | | |
| software, algorithm (Imaris (Bitplane, Version 8.1)) | Imaris | Bitplane | | |
| software, algorithm (Fiji software) | Fiji | DOI: 10.1038/nmeth.2019 | | |
| software, algorithm (Adobe Bridge and Photoshop (Adobe Systems)) | Adobe Bridge and Photoshop | Adobe | | |
| software, algorithm (Zen 8.1 Software (Zeiss)) | Zen | Zeiss | | |
| software, algorithm (Excel 2010 (Microsoft Office)) | Excel | Microsoft | | |
| software, algorithm (PikoReal software 2.2 (ThermoFisher Scientific)) | PikoReal software | ThermoFisher Scientific | | |

## Zebrafish genetics and maintenance

Handling of zebrafish was done in compliance with German and Brandenburg state law, carefully monitored by the local authority for animal protection (LUVG, Brandenburg, Germany; Animal protocol #2347-18-2015). The following strains of zebrafish were maintained under standard conditions as previously described (*Westerfield et al., 1997*): krit1$^{ty219c}$ (*Mably et al., 2006*), ccm2$^{m201}$ (*Driever et al., 1996*), Tg(EPV.Tp1-Mmu.Hbb:Venus-Mmu.Odc1)$^{s940}$ [here referred to as *Tg(TP1: VenusPEST)$^{s940}$*] (*Ninov et al., 2012*), Tg(kdrl:EGFP)$^{s843}$ (*Jin et al., 2005*), Tg(fli1a:GAL4FF)$^{ubs3}$ (Herwig et al., 2011). Some embryos were treated with 1-phenyl-2-thiourea (PTU) (Sigma Aldrich) prior to the appearance of pigmentation.

## Morpholino injections

The following morpholinos were used: *tnnt2a* (5'-CATGTTTGCTCTGATCTGACACGCA-3') (2 ng/ embryo) (*Sehnert et al., 2002*), *klf2a* (5'-CTCGCCTATGAAAGAAGAGAGGATT-3') (1 ng/embryo) (*Nicoli et al., 2010*), *klf2b* (5'-AAAGGCAAGGTAAAGCCATGTCCAC-3') (5 ng/embryo) (*Renz et al., 2015*), *heg1* (5'-GTAATCGTACTTGCAGCAGGTGACA-3') (5 ng/embryo) (*Mably et al., 2003*).

## Molecular cloning

The open reading frames of zebrafish *krit1* (NM_001317001), *klf2a* (NM_131856), and *ccm2* (NM_001002315) were amplified by PCR and cloned into the Gateway pDONR221 vector (referred to as pME-*krit1*, pME-*klf2a*, and pME-*ccm2*, respectively). To generate the *krit1* fusion plasmids, *EGFP* was fused in the N-terminal site of *krit1* (pME-*EGFP-krit1*).

These middle entry constructs were used in combination with the following Gateway plasmids to generate final constructs by standard Gateway cloning recombination reactions: pDestTol2 (#426, obtained from N. Lawson), p5E-*hsp70l* (#222, obtained from N. Lawson), p5E-*CMV/SP6* (#382, obtained from Chien, Univ. Utah), p5E-*UAS* (#327, obtained from Chien, Univ. Utah), p3E-*IRE-S_EGFPpA* (#389, obtained from N. Lawson), p3E-*cryaa:EGFPpA*, p3E-*EGFPpA* (#366, obtained from Chien, Univ. Utah), p3E-*pA* (#383, obtained from Chien, Univ. Utah).

Final plasmids:

pDest_Tol2pA_*Hsp70l:Krit1_IRES_EGFPpA*,

pDest_Tol2pA_*Hsp70l:Ccm2_IRES_EGFPpA*,

pDest_Tol2pA_*Hsp70l:Klf2a_IRES_EGFPpA*,

pDest_Tol2pA_*UAS:EGFP-Krit1pA,cryaa:EGFPpA*,

pDest_Tol2pA_*CMV/SP6:EGFP-Krit1pA*.

## Generation of transgenic lines of zebrafish

Transformation plasmids (25 pg/embryo) were co-injected together with mRNA encoding Tol2 transposase (50 pg/embryo) into one-cell-stage zebrafish embryos. Several independent transgenic lines were established for each construct. In functional tests and localization studies, these independent lines resulted in comparable phenotypes. One transgene for each construct was selected for further analyses:

Tg(hsp70l:Krit1_IRES_EGFP)^md6

Tg(hsp70l:Ccm2_IRES_EGFP)^md12

Tg(hsp70l:Klf2a_IRES_EGFP)^pbb22

*Tg(UAS:EGFP-Krit1,cryaa:EGFP)^pbb21* [here referred to as *Tg(UAS:EGFP-Krit1)^pbb21*]

## Heat-shock experiments

To assess levels of *klf2a* mRNA following *krit1* overexpression, *Tg(hsp70l:Krit1_IRES_EGFP)^md6* was crossed to wild-type and the resulting embryos were heat-shocked at 14 hpf (30 min at 37°C), at 24 hpf (40 min at 38°C), and at 40 hpf (45 min at 38°C). Alternatively, for the experiment shown in **Figure 2C,D**, embryos obtained from *Tg(hsp70l:Krit1_IRES_EGFP)^md6* or *Tg(hsp70l:Ccm2_IRES_EGFP)^md12* crossed to *Tg(TP1:VenusPEST)^s940* were additionally heat-shocked at 48 hpf (45 min at 38°C). For the experiment shown in **Figure 1B**, *Tg(hsp70l:Klf2a_IRES_EGFP)^pbb22* were crossed to wild-type and the resulting embryos were heat-shocked at 48 hpf (45 min at 38°C).

## mRNA injection experiments

Capped mRNA encoding EGFP-Krit1 or Heg1 was synthesized using SP6 polymerase (mMessage Machine kit, Ambion). For rescue experiments, 150 pg of *egfp-krit1* mRNA was injected into one-cell-stage zebrafish embryos. Embryos were selected for EGFP fluorescence at 6 hpf and the genotype was assessed by sequencing. For *heg1* overexpression, 100 pg of *heg1* mRNA was injected into one-cell-stage zebrafish embryos.

## Statistical analysis of the efficiency of mRNA rescue experiments

Statistical analysis of the efficiency of *egfp-krit1* mRNA rescue experiments (as shown in **Figure 3—figure supplement 1**) was done using GraphPad Prism6 (Student's t-test, p=0.008 and p=0.005, respectively).

| | Unpaired t-test | Mean diff. | Summary | Individual P value |
|---|---|---|---|---|
| 48 hpf *egfp-krit1* mRNA | WT vs. *krit1* | 100.0 | **** | <0.0001 |
| | WT vs. WT + *egfp-krit1* mRNA | 0.0 | ns | >0.9999 |
| | WT vs. *krit1* + *egfp-krit1* mRNA | 28 | *** | 0.0004 |

Statistical analysis of the rescue efficiency of *egfp-krit1* mRNA injection into *krit1*[ty219c] mutant embryos was based on the presence of blood-flow at 48 hpf for all embryos, and at 96 hpf for those embryos that had blood-flow at 48hpf. The percentages of embryos with blood-flow was recorded for three individual experiments and compared with GraphPad Prism6, using a 2way ANOVA with Multiple comparisons without correction.

| 48 hpf | Number of WT with blood-flow/total | Number of *krit1*[ty219c] with blood-flow/total | Number of WT + *egfp-krit1* mRNA with blood-flow/total | Number of *krit1*[ty219c] + *egfp-krit1* mRNA with blood-flow/total |
|---|---|---|---|---|
| 1 | 5/5 | 0/3 | 32/32 | 15/15 |
| 2 | 78/78 | 0/17 | 40/40 | 42/42 |
| 3 | 69/69 | 0/26 | 21/21 | 11/11 |

| 96 hpf | Number of WT with blood-flow at 48 hpf and 96 hpf/total | Number of *krit1*[ty219c] with blood-flow at 48 hpf and 96 hpf/total | Number of WT + *egfp-krit1* mRNA with blood-flow at 48 hpf and 96 hpf/total | Number of *krit1*[ty219c] + *egfp-krit1* mRNA with blood-flow at 48 hpf and 96 hpf/total |
|---|---|---|---|---|
| 1 | 67/67 | 0/28 | 59/59 | 5/22 |
| 2 | 70/70 | 0/11 | 20/21 | 1/11 |
| 3 | 57/57 | 0/13 | 30/30 | 7/18 |

| | Within each row, compare columns (simple effects within rows) | | | |
|---|---|---|---|---|
| | Uncorrected Fisher's LSD | Mean diff. | Summary | Individual P value |
| 48 hpf | WT vs. *krit1* | 100.0 | **** | <0.0001 |
| | WT vs. WT + *egfp-krit1* mRNA | 0.0 | ns | >0.9999 |
| | WT vs. *krit1* + *egfp-krit1* mRNA | 0.0 | ns | >0.9999 |
| 96 hpf | WT vs. *krit1* | 100.0 | **** | <0.0001 |
| | WT vs. WT + *egfp-krit1* mRNA | 1.590 | ns | 0.7212 |
| | WT vs. *krit1* + *egfp-krit1* mRNA | 76.44 | **** | <0.0001 |

## Quantifications of ventricular endocardial cell numbers

Endocardial cell numbers of the ventricles at 48 hpf and 55 hpf of WT and *krit1*[ty219c] mutants were quantified as previously shown (*Renz et al., 2015*). Nuclei were visualized by *Tg(kdrl:GFP)*[s843] expression and were counted within the ventricle (for endocardium). Cell numbers are shown as means with SEM. Prism 6 (GraphPad) was used to perform an unpaired t-test (*Figure 3—figure supplement 2*). Means are significantly different when p<0.05.

| 48 hpf | | n | Average cell number | SEM | p-value |
|---|---|---|---|---|---|
| | WT | 6 | 68 | ±2.362 | 0.5723 |
| | *krit1* + *egfp-krit1* mRNA | 6 | 65 | ±3.572 | |

| 55 hpf | | n | Average cell number | SEM | p-value |
|---|---|---|---|---|---|
| | WT | 3 | 78 | ±2.028 | 0.0093 |
| | *krit1* + *egfp-krit1* mRNA | 3 | 112 | ±7.024 | |

## Whole-mount immunohistochemistry and *in situ* hybridization

Zebrafish whole-mount immunohistochemistry was performed on 30 hpf, 48 hpf, 54 hpf, and 96 hpf embryos as previously described (*Renz et al., 2015*). The following antibodies were used: rabbit anti-VE-Cadherin (Cdh5) (1:200, a kind donation from Markus Affolter, Basel) (*Blum et al., 2008*), mouse anti-Zn-8/Alcam (1:25, Developmental Studies Hybridoma Bank), and mouse anti-Myh6 (1:10, Developmental Studies Hybridoma Bank, S46). Secondary antibodies were Alexa Fluor 633-conjugated goat anti-rabbit (1:200, Invitrogen A21070), Rhodamine Red-X-conjugated goat anti-mouse (1:200, Jackson ImmunoResearch Laboratories 115-295-003), and Dylight 649-conjugated goat anti-mouse (1:200, Jackson ImmunoResearch Laboratories 115-495-003). Rhodamine-Phalloidin (1:250, Invitrogen R415) was incubated together with secondary antibodies. For Cdh5 antibody staining, embryos were fixed with 2% PFA overnight and permeabilized with PBST with 0.5% Triton X-100 for 1 hr and subsequently incubated with primary antibody diluted in PBST with 0.2% Triton X-100, 1% BSA, and 5% NGS. All specimens were mounted in SlowFade Gold (Invitrogen S36936). Images were recorded on LSM 710, or LSM 780 confocal microscopes (Zeiss) and processed with Imaris (Bitplane, Version 8.1) or Fiji software (*Schindelin et al., 2012*).

Whole-mount *in situ* hybridization experiments were performed as previously described (*Jowett and Lettice, 1994*). For all experiments embryos were fixed overnight with 4% PFA. For *Figures 3G–L*, 54 hpf and 96 hpf embryos were stained with a *klf2a* probe previously published (*Renz et al., 2015*). For *Figures 4A–B*, 48 hpf embryos were stained with a *notch1b* probe previously published (*Walsh and Stainier, 2001*) (the plasmid was a kind gift from Didier Stainier). For *Figure 1—figure supplements 1*, 30 hpf and 48 hpf embryos were stained with a *heg1* probe previously published (*Münch et al., 2017*). For *Figure 4—figure supplements 1C–D*, 48 hpf embryos were stained with a *tbx2b* probe (*Sedletcaia and Evans, 2011*). Images were recorded with 10x or 20x objectives on an Axioskop (Zeiss) with an EOS 5 D Mark III (Canon) camera, and processed using Adobe Bridge and Photoshop (Adobe Systems).

## Live imaging

Embryos were dechorionated manually and embedded in 1% low melting agarose (Lonza 50081) containing 0.16 mg/ml Tricaine (3-amino benzoic acidethylester, Sigma-Aldrich A-5040). Images were recorded with a LSM 710 confocal microscope (Zeiss) at 10x.

## Ratiometric corrected total tissue fluorescence (CTTF) image analysis

The ratiometric measurements of $Tg(TP1:VenusPEST)^{s940}$ fluorescence in atrium versus ventricle and AVC of the embryonic heart (*Figure 2E*) was done as previously described (*McCloy et al., 2014*). As overexpression of *krit1* also resulted in an upregulation of EGFP, the overlap of EGFP with $Tg(TP1:VenusPEST)^{s940}$ fluorescence was safely separated into two different channels by recording the images using the online fingerprinting function of the Zen 8.1 Software (Zeiss). Images were acquired maintaining fixed recording settings. From a 3D confocal image, the z-portion corresponding to the entire heart was selected. Regions of single z-planes corresponding to atrium or to ventricle plus the AVC were selected and fluorescence was measured until entire hearts were covered. The corrected total tissue fluorescence [CTTF = integrated density – (area of selected tissue x mean fluorescence of background readings)] was calculated using Excel 2010 (Microsoft Office). Next, all CTTF values collected for atrium as well as for ventricle plus AVC of each heart were averaged. The mean of atrium CTTF values was divided by the mean of ventricle plus AVC CTTF values. A mean value of 1 corresponds to an equal expression of $Tg(TP1:VenusPEST)^{s940}$ in both heart chambers. All measurements and tissue selection were performed using Fiji software (*Schindelin et al., 2012*). Statistical analysis was done with Excel 2010 (Microsoft Office) using an unpaired t-test (n: number of hearts analyzed, *hs*: heat-shock, *p<0.05, **p<0.01).

| | n | Ratio *TP1:VenusPEST* Atrium/Ventricle (CTTF) | SEM | p |
|---|---|---|---|---|
| WT | 19 | 0.213 | ±0.024 | 0.0005 |
| *hs*:Krit1 | 18 | 0.418 | ±0.048 | |

| | n | Ratio *TP1:VenusPEST* Atrium/Ventricle (CTTF) | SEM | p |
|---|---|---|---|---|
| WT | 19 | 0.290 | ±0.040 | 0.0261 |
| *hs*:Ccm2 | 20 | 0.428 | ±0.044 | |

To quantify the fluorescent levels of *Tg(TP1:VenusPEST)*$^{s940}$ in the AVC of WT (heat-shock control) (*Figure 2F*), *hs*:Krit1 and *hs*:Ccm2, a section of 20 µm corresponding to the entire AVC was selected from a maximum projection of confocal z-stacks of the AVC. The CTTF value of each AVC was measured and all the values were divided by the average CTTF of the controls for normalization.

| | n | CTTF mean | SEM | p |
|---|---|---|---|---|
| WT | 19 | 1.000 | ±0.121 | 0.0426 |
| *hs*:Krit1 | 19 | 0.671 | ±0.099 | |

| | n | CTTF mean | SEM | p |
|---|---|---|---|---|
| WT | 19 | 1.000 | ±0.059 | 0.0481 |
| *hs*:Ccm2 | 20 | 0.798 | ±0.078 | |

To quantify the fluorescent levels of EGFP-Krit1 in WT and *heg1* morphants, maximum projections of confocal z-stacks of manually extracted 30 hpf hearts (*Figure 1C–E*) or of the caudal plexus region (*Figure 1—figure supplement 2*) were used. The background intensity was measured in five different areas of each image. To measure the caudal plexus region, an area between two intersegmental vessels from the most dorsal to the most ventral side of the caudal plexus was selected. This was repeated in three different areas for each embryo. All measurements and tissue selection were performed using Fiji software (*Schindelin et al., 2012*). The CTTF was calculated using Excel 2010 (Microsoft Office). All the values were divided by the average intensity of the controls.

| | n | CTTF mean hearts | SEM | P |
|---|---|---|---|---|
| EGFP-Krit1 levels WT | 5 | 1.000 | ±0.074 | ≤0.0005 |
| EGFP-Krit1 levels *heg1* MO | 10 | 0.600 | ±0.049 | |

| | n | CTTF mean caudal plexus region | SEM | p |
|---|---|---|---|---|
| EGFP-Krit1 levels WT | 5 | 1.000 | ±0.081 | <0.0001 |
| EGFP-Krit1 levels *heg1* MO | 5 | 0.305 | ±0.059 | |

## Quantifications of mRNA expression by RT-qPCR

For RT-qPCR experiments, 25 zebrafish embryos were pooled for each condition (three biological replicates). For heat-shock conditions, controls of each biological replicate were composed of heat-shocked siblings from the same clutch lacking EGFP expression. Total RNA was extracted with Trizol (Sigma) and Phase Lock Gel Heavy tubes (1.5 mL, Prime 5) and the corresponding cDNA was synthesized from total RNA with the RevertAid H Minus First Strand cDNA Synthesis kit (ThermoFisher Scientific). RT-qPCR experiments were performed as described (*Renz et al., 2015*) using 18 ng cDNA per technical replicate and the KAPA Sybr Fast qPCR kit (Peqlab) on a PikoReal 96 Real-Time PCR System (ThermoFisher Scientific). Cycle threshold (Ct) values were determined by PikoReal software 2.2 (ThermoFisher Scientific). *eif1b* was used as a housekeeping gene.

The following primers were used for qRT-PCR:

| Target | Sequence 5'–3' |
|---|---|
| *heg1* | Fw _ GCTCTTATTGTCACCTGCTGC<br>Rv _ CGGATAGATGCAGGAATGCC |
| *krit1* | Fw _ GTCTGAGCACTAGTGAGGGTG<br>Rv _ GACCTGTCCTGTGAAAAACGC |
| *klf2a* | Fw _ CTGGGAGAACAGGTGGAAGGA<br>Rv _ CCAGTATAAACTCCAGATCCAGG |
| *eif1b* | Fw _ CAGAACCTCCAGTCCTTTGATC<br>Rv _ GCAGGCAAATTTCTTTTTGAAGGC |

Results were analyzed using the comparative threshold cycle method (2–ΔΔCt) to compare gene expression levels between samples as previously described (*Livak and Schmittgen, 2001*). As an internal reference, we used zebrafish *eif1b* (*Renz et al., 2015*). Control sample values were normalized to 1. In the table below, the mean of the fold changes and the corresponding SEM of each biological replicate after normalization is shown. As each single biological replicate represents an independent experiment from an independent clutch of embryos, ratio paired t-tests were done with Prism 6 (GraphPad) (Stg: developmental stage of zebrafish embryos at RNA extraction, n: number of independent biological replicates analyzed, *p<0.05, **p<0.01, hs: heat-shock, mean >1 corresponds with target upregulation; mean <1 represents target downregulation).

| Treatment | Stg (hpf) | n | heg1 | | | krit1 | | | klf2a | | |
|---|---|---|---|---|---|---|---|---|---|---|---|
| | | | Mean | SEM | p | Mean | SEM | p | Mean | SEM | p |
| *tnnt2a* MO | 54 | 3 | 0.85 | 0.008 | 0.013 | 1.06 | 0.009 | 0.103 | - | - | - |
| *klf2a/b* MO | 54 | 4 | 0.77 | 0.034 | 0.045 | 1.17 | 0.029 | 0.103 | - | - | - |
| *hs*:Klf2a | 54 | 3 | 2.56 | 0.089 | 0.044 | 1.38 | 0.061 | 0.151 | - | - | - |
| *hs*:Krit1 | 48 | 4 | - | - | - | 9.77 | 0.063 | 0.001 | 0.63 | 0.059 | 0.044 |
| *heg1* mRNA | 24 | 3 | 4.01 | 0.059 | 0.0095 | - | - | - | 0.31 | 0.107 | 0.04 |
| *heg1* mRNA | 48 | 3 | 2.47 | 0.036 | 0.0082 | - | - | - | 0.68 | 0.023 | 0.019 |
| *egfp-krit1* mRNA | 24 | 3 | - | - | - | 7.06 | 0.125 | 0.021 | - | - | - |
| *egfp-krit1* mRNA | 48 | 3 | - | - | - | 3.24 | 0.053 | 0.011 | - | - | - |
| *egfp-krit1* mRNA | 96 | 3 | - | - | - | 1.20 | 0.161 | 0.675 | - | - | - |

To prove that differences in average values of *krit1* mRNA levels between 24 hpf and 48 hpf, 48 hpf and 96 hpf, 24 hpf and 96 hpf were significant, a ratio paired t-test was performed, as in each single biological replicate, embryos of 24, 48, and 96 hpf were from the same clutches. The mean of the ratio indicated in the table below is always between 1 and 0, and this accounts for target downregulation.

| Compared stages (hpf) | Mean of the ratio | SEM | p |
|---|---|---|---|
| 24–48 | 0.4594 | ±0.082 | 0.027 |
| 48–96 | 0.3691 | ±0.110 | 0.029 |
| 24–96 | 0.1696 | ±0107 | 0.009 |

## Acknowledgements

We would like to thank M Affolter and H Belting (Basel), and D Stainier (Bad Nauheim) for reagents or fish lines. Thanks to A Michels for contributing some of the RT-qPCR data. Thanks also to O Baumann, A Hubig, M Kneiseler, A Kühnel, and B Wuntke for technical support. For critical reading and discussions of the project and the manuscript we are indebted to C Albiges-Rizo, E Faurobert, R

Hodge, and team members. The group has been generously supported by the excellence cluster REBIRTH, SFB958, and by Deutsche Forschungsgemeinschaft (DFG) projects SE2016/7-2 and SE2016/10-1 to SA-S.

## Additional information

### Funding

| Funder | Grant reference number | Author |
|---|---|---|
| Deutsche Forschungsge-meinschaft | Excellence Cluster REBIRTH | Stefan Donat |
| Deutsche Forschungsge-meinschaft | SFB 958 | Cécile Otten |
| Deutsche Forschungsge-meinschaft | Project number SE2016/7-2 | Alessio Paolini<br>Cécile Otten<br>Marc Renz |
| Deutsche Forschungsge-meinschaft | Project number SE2016/10-1 | Alessio Paolini<br>Cécile Otten<br>Marc Renz |

The funders had no role in study design, data collection and interpretation, or the decision to submit the work for publication.

### Author contributions
Stefan Donat, Marta Lourenço, Alessio Paolini, Conceptualization, Resources, Data curation, Software, Formal analysis, Validation, Investigation, Visualization, Methodology, Writing—original draft, Writing—review and editing; Cécile Otten, Conceptualization, Resources, Data curation, Software, Formal analysis, Validation, Investigation, Visualization, Methodology, Writing—review and editing; Marc Renz, Resources, Methodology; Salim Abdelilah-Seyfried, Conceptualization, Data curation, Formal analysis, Supervision, Funding acquisition, Visualization, Methodology, Writing—original draft, Project administration, Writing—review and editing

### Author ORCIDs
Stefan Donat (iD) http://orcid.org/0000-0003-3901-3733
Alessio Paolini (iD) http://orcid.org/0000-0001-7002-7303
Cécile Otten (iD) http://orcid.org/0000-0002-8230-7350
Salim Abdelilah-Seyfried (iD) http://orcid.org/0000-0003-3183-3841

### Ethics
Animal experimentation: Handling of zebrafish was done in compliance with German and Brandenburg State law, carefully monitored by the local authority for animal protection (LUGV, Brandenburg, Germany; Animal protocol#2347-18-2015 ).

### Decision letter and Author response
Decision letter https://doi.org/10.7554/eLife.28939.018
Author response https://doi.org/10.7554/eLife.28939.019

## Additional files

### Supplementary files
• Transparent reporting form
DOI: https://doi.org/10.7554/eLife.28939.016

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
