## [Decision Letter]

Thank you for submitting your article "CCM proteins control endocardial mechanosensitivity during zebrafish valvulogenesis" for consideration by *eLife*. Your article has been reviewed by three peer reviewers, and the evaluation has been overseen by a Reviewing Editor and Marianne Bronner as the Senior Editor. The following individuals involved in review of your submission have agreed to reveal their identity: Ian C Scott (Reviewer #1); Dimitris Beis (Reviewer #3).

The reviewers have discussed the reviews with one another and the Reviewing Editor has drafted this decision to help you prepare a revised submission.

Summary:

In this manuscript, Donat and colleagues investigate the roles of the CCM-associated genes *heg1* and *krit1* during atrioventricular valve leaflet formation in the endocardium of the developing zebrafish heart. The authors first show that *heg1* transcripts are positively regulated by blood flow. Heg1 in turn acts to stabilize Krit1, and these genes repress *klf2a*, a major mechanosensitive gene. As a result, the absence of Heg1 and Krit1 function leads to overexpression of *klf2a* and mis-activation of Notch signaling throughout the endocardium, which impairs valve leaflet formation. Overall, the authors present a model where Heg1/Krit1/CCM2 "establishes a negative feedback loop that desensitizes endothelial cells to blood flow-induced mechanosensitive signaling by dampening expression levels of Klf2a". As the links between Heg/Krit1/CCM2 and Klf2a/Notch have been previously described (including by Dr. Abdelilah-Seyfried), the most novel element of this model is the implication of the Heg1/Krit1/CCM2 pathway in valve leaflet development, with its potential function being a means to fine-tune the response to flow during valve development. This is an interesting model, particularly in terms of providing a mechanistic link between blood flow and specific features of cardiac morphology, and it would therefore be of interest to the readers of *eLife*. However, there are a number of caveats inherent in the authors' experimental design and results, and further work is needed to fully support the authors' interpretations and conclusions.

Essential revisions:

1) Most of the expression data is supported by quantitative PCR (i.e. Figure 1, Figure 2) on vascular endothelial cells. This technique, although quantitative, does not provide any information about the localization of the expression of the studied genes. Whole mount in situ hybridizations would greatly help to reveal spatial differences in the expression of these genes, particularly in the heart. Ideally, qPCR should be performed in isolated AV endocardial cells, given the focus of the manuscript.

2) Figure 1: The stabilization of Krit1 protein by Heg1 is shown in trunk vessels, yet this research is focused on valve development. This should be analyzed in the AVC. Additionally, the conclusion that Heg1 stabilizes the levels of Krit1 is based on an experiment using a Gal4-UAS system that overexpresses EGFP-Krit1. Thus, it is somewhat challenging to draw a meaningful conclusion about the ability of Heg1 to stabilize Krit1 protein levels from these non-physiologic studies. Western analysis would be a better approach in determining Krit1 levels in *heg1* morphant and wild-type hearts.

3) Figure 2: The evidence that "endocardial cells become desensitized to blood flow-induced mechanosensitive signaling when *krit1* or *ccm2* are overexpressed" is not compelling. The authors state that hearts with CCM1/2 overexpression resemble silent heart morphant (no flow) hearts, and that the atrium/ventricle flt1:YFP expression ratio is increased over time, but this may simply reflect aberrant heart development rather than a specific desensitization. In addition, the authors' use of flt1:YFP as a reporter of flow responses is based on the prior studies of Hogan, which were focused on the vascular endothelium. However, the ability of this line to sense hemodynamic flow has not been validated in the heart. Particularly concerning is the expression of flt1:YFP in the ventricle of the *tnnt2a* MO hearts. Is this endocardial or myocardial expression that is observed? Using endocardial and myocardial lines in combination with the flt1:YFP might help resolve these issues. Performing studies to validate the ability of this line to sense hemodynamic flow in the heart would help with interpretation of studies as well.

4) Although the authors conclude that *notch1b* is expressed throughout the endocardium in the *krit1* mutant, the *notch1b* in situs and the notch reporter do not support this interpretation. It is possible that the AVC region is expanded in the *krit1* mutant. How are the authors defining the AVC? Without atrial, ventricular or AV boundary markers, it is difficult to determine where the endocardial cushions are in the *krit1* mutant. Providing this data would help with interpretation of the studies. In addition, it would be useful to examine the *ccm2* mutant in addition to the *krit1* mutant, to strengthen the demonstration of how loss of ccm genes affects endocardial notch activation.

5) The studies on the overexpression of exogenous mRNA of egfp-*krit1* in the *krit1* mutant do not have sufficient controls and validation of assays to allow for a meaningful interpretation. Many assumptions are made in order for the authors to come to their conclusions. For example, there is insufficient evidence to show that injected mRNA is no longer present at 55 or 96 hpf. Moreover, over/mis-expression of *krit1* may lead to a phenotype irrespective of the *krit1* mutant. In fact, the phenotype is surprisingly similar to that found in late Notch inhibition and in the *klf2a* mutant (Beis et al., 2005, Steed et al., 2016).

6) It would be beneficial to provide further support for the requirement of Krit1 for the generation of abluminal cell fates during valvulogenesis. The transient rescue of *krit1* mutants allows AVC endocardial cells to differentiate properly at 48 hpf. Alcam levels do appear higher in most AVC cells at 96 hpf (Figure 3, duplicated in Figure 3—figure supplement 4’) but the morphology of the AV valve seems to also include abluminal cells (cardiac jelly cells or mesenchymal-like cells). Repeating the experiment in combination with a Wnt signaling readout (TCF reporter or other) and/or using the inducible *hs*:Krit1 line in the *krit1* mutant background to better control *krit1* levels, would strengthen their claim.

7) The model in Figure 5 is somewhat confusing. It might be helpful to show a model for the *ccm* mutant with blood flow and then when there is no blood flow. Based on the authors' current model, it seems that Notch would be activated throughout the endocardium in the *ccm* mutant with blood flow since *ccm* is not present to block *klf2a* expression. If this is the case, this seems to be in conflict with the current *ccm* mutant and no blood flow model where the lack of blood flow would lead to no *klf2a* expression and downstream Notch signaling. Additionally, the authors also speculate "Notch-mediated lateral inhibition" in their model; however, there is insufficient data to support this notion. This should be modified in the model.

[Editors' note: further revisions were requested prior to acceptance, as described below.]

Thank you for resubmitting the revised version of your manuscript entitled "CCM proteins control endocardial mechanosensitivity during zebrafish valvulogenesis" for further consideration at *eLife*. Your revised manuscript has now been favorably evaluated by Marianne Bronner (Senior editor), a Reviewing editor, and its original three reviewers.

The reviewers appreciate the improvements made in your revised manuscript, but they have also raised several issues that remain to be addressed through further revision, as outlined below.

1) The Abstract is confusing in its current form, as it is not made clear in which direction many of the genes/pathways mentioned affect outcome. As an example: does the abstract make it clear whether *heg1* expression is increased or reduced by low blood flow? The last paragraph of the Introduction section is similarly vague and does not refer to the *heg1* results.

2) In the Title, it may be better to mention CCM1/2, since CCM3 is not addressed in this study. Non-CCM experts may not appreciate what "CCM proteins" are.

3) From the data shown in the revised manuscript, it is not clear whether *heg1* is expressed in the endocardium or the myocardium, although it is stated that it is expressed in the endocardium. Can the evidence for this be clarified?

4) The use of the Notch reporter, in place of the previous Flt reporter, to monitor flow response is appreciated. However, the images in Figure 2 are hard to interpret. Although the ratios measured by the authors (Figure 2) demonstrate the same trend following overexpression of Krit1 and Ccm2, the patterns of expression shown in Figure 2 seem markedly different; hard to interpret. This apparent discrepancy should be addressed. In addition, in light of recent studies showing myocardial Notch reporter activity in developing zebrafish hearts, it is unclear whether the Notch reporter activity shown here (and in Figure 4) is endocardial or myocardial. Can this be clarified?

5) The authors utilize Cdh5 as a marker for the lumenal side of the AV valve, but they do not cite references for its use as a lumenal marker. If they are introducing this marker here, its characterization as such should be clearly described.

6) In Figure 4—figure supplement 1, the authors use a single marker (Myh6) to define the location of the AVC in *krit1* mutants. This marker indicates the edge of the atrium but isn't sufficient to demonstrate whether the AVC is or isn't expanded in the mutant hearts. Can the authors use specific AVC markers to indicate whether the AVC region is expanded in *krit1* mutants?

7) At the beginning of the Discussion section, the authors allude to evidence that Cdh5/VEcadherin plays into this pathway, but such evidence is not reported in this manuscript.

8) The new model figure is appreciated, but, in light of what is suggested by Figure 1, it does not address the effect of flow on Heg1 and Krit1 levels. This is confusing – can it be revised to include these elements of the model?

---

## [Author Response]

Essential revisions:1) Most of the expression data is supported by quantitative PCR (i.e. Figure 1, Figure 2) on vascular endothelial cells. This technique, although quantitative, does not provide any information about the localization of the expression of the studied genes. Whole mount in situ hybridizations would greatly help to reveal spatial differences in the expression of these genes, particularly in the heart. Ideally, qPCR should be performed in isolated AV endocardial cells, given the focus of the manuscript.

Within the revised version of the manuscript, we now provide expression data of *heg1* mRNA (Figure 1). Consistent with a blood-flow responsive expression, *heg1* mRNA has a stronger expression at the AVC and in the ventricle at 48 hpf. In support of our finding, another study reported a similar pattern of *heg1* mRNA cardiac expression recently (Münch et al., 2017). We are now also referring to this work in our revised manuscript. The expression of *krit1* has been published previously (Mably et al.,2006) and, similar to that study, we have not found any regional expression of *krit1* that would point at a blood-flow responsive regulation. This finding is in tune with the fact that *krit1* mRNA levels are not affected in *tnnt2a* morphants as assayed by RT-qPCR. This lends further support to an important role particularly of *heg1* in response to biomechanical stimuli. The reviewers’ suggestion to use isolated AVC endocardial cells for RT-qPCR experiments is very good but technically very challenging. Unfortunately, we have not been able yet to establish a technology for AVC dissection.

2) Figure 1: The stabilization of Krit1 protein by Heg1 is shown in trunk vessels, yet this research is focused on valve development. This should be analyzed in the AVC. Additionally, the conclusion that Heg1 stabilizes the levels of Krit1 is based on an experiment using a Gal4-UAS system that overexpresses EGFP-Krit1. Thus, it is somewhat challenging to draw a meaningful conclusion about the ability of Heg1 to stabilize Krit1 protein levels from these non-physiologic studies. Western analysis would be a better approach in determining Krit1 levels in heg1 morphant and wild-type hearts.

Within revised Figure 1, we now show that EGFP-Krit1 levels are significantly reduced within the heart. This finding is in tune with the changes of EGFP-Krit1 levels within the vasculature (now shown in Figure 1—figure supplement 2).

Due to the lack of an antibody that recognizes zebrafish Krit1, it has remained a challenge to quantify the physiological levels of Krit1 upon loss of Heg1. To assess physiological levels of Krit1, we therefore undertook mass spectrometric analyses of the cardiac proteome under WT and *heg1* mutant conditions (with 150 purified hearts per genetic condition and replicate) in an attempt to detect CCM complex proteins. Unfortunately, this approach was not sufficiently sensitive to detect Krit1 protein levels.

Faurobert and colleagues recently demonstrated that, similar to our findings, in HUVECs, a loss of CCM2 also causes a depletion of KRIT1 (Faurobert et al., (2013).

3) Figure 2: The evidence that "endocardial cells become desensitized to blood flow-induced mechanosensitive signaling when krit1 or ccm2 are overexpressed" is not compelling. The authors state that hearts with CCM1/2 overexpression resemble silent heart morphant (no flow) hearts, and that the atrium/ventricle flt1:YFP expression ratio is increased over time, but this may simply reflect aberrant heart development rather than a specific desensitization. In addition, the authors' use of flt1:YFP as a reporter of flow responses is based on the prior studies of Hogan, which were focused on the vascular endothelium. However, the ability of this line to sense hemodynamic flow has not been validated in the heart. Particularly concerning is the expression of flt1:YFP in the ventricle of the tnnt2a MO hearts. Is this endocardial or myocardial expression that is observed? Using endocardial and myocardial lines in combination with the flt1:YFP might help resolve these issues. Performing studies to validate the ability of this line to sense hemodynamic flow in the heart would help with interpretation of studies as well.

We agree with the reviewers that the ability of the Tg(flt1:YFP) line to sense the hemodynamic forces of blood flow has not previously been shown. Within the revised manuscript, we have now repeated the above experiments using the well-established blood flow-sensitive Notch reporter line *Tg(EPV.Tp1-Mmu.Hbb:Venus-Mmu.Odc1)^s940^*(Ninov et al., 2012). Within the revised manuscript, we now show that the forced overexpression of Krit1 or Ccm2 alters the expression of the *Tg(EPV.Tp1-Mmu.Hbb:Venus-Mmu.Odc1)^s940^* reporter within the endocardium which, in WT, is particularly strong within the ventricle and AVC. Due to the overexpression of Krit1 or Ccm2, the expression from the *Tg(TP1:VenusPEST)^s940^* reporter is weakened and not restricted to the ventricle and AVC region at 54 hpf (Figure 2). These expression changes of the Notch reporter line occur while blood flow is not obviously affected as assayed by visual inspection. Hence, we suggest that the blood flow responsiveness of endocardial cells is affected.

4) Although the authors conclude that notch1b is expressed throughout the endocardium in the krit1 mutant, the notch1b in situs and the notch reporter do not support this interpretation. It is possible that the AVC region is expanded in the krit1 mutant. How are the authors defining the AVC? Without atrial, ventricular or AV boundary markers, it is difficult to determine where the endocardial cushions are in the krit1 mutant. Providing this data would help with interpretation of the studies. In addition, it would be useful to examine the ccm2 mutant in addition to the krit1 mutant, to strengthen the demonstration of how loss of ccm genes affects endocardial notch activation.

To visualize the AVC region for a better assessment of the AVC endocardial Notch expression, we have counterstained the hearts using chamber-specific markers in WT and *krit1* mutants (Figure 4—figure supplement 1). The fact that myocardial chamber identities are not affected in *ccm* mutants provides a strong boundary-marker for the AVC. Based on that sharply defined border between both chambers, we have assessed the endocardial Notch reporter expression which expands far beyond that boundary region.

We also agree with the reviewers that it is important to verify the results obtained in *krit1* mutants also in *ccm2* mutants. The appropriate results for these mutants are now shown for *ccm2* (Figure 4’) and for *krit1* mutants (Figure 4—figure supplement 1). In both cases, the domain of Notch signaling is expanded within the endocardium.

5) The studies on the overexpression of exogenous mRNA of egfp-krit1 in the krit1 mutant do not have sufficient controls and validation of assays to allow for a meaningful interpretation. Many assumptions are made in order for the authors to come to their conclusions. For example, there is insufficient evidence to show that injected mRNA is no longer present at 55 or 96 hpf. Moreover, over/mis-expression of krit1 may lead to a phenotype irrespective of the krit1 mutant. In fact, the phenotype is surprisingly similar to that found in late Notch inhibition and in the klf2a mutant (Beis et al., 2005, Steed et al., 2016).

We have now measured *krit1* mRNA levels over time and find that the levels are depleted by 96 hpf (shown in revised Figure 3—figure supplement 3). Since there is no evidence for EGFP-Krit1 protein levels at that stage (no EGFP detectable), the evidence is very strong that we are analyzing a *krit1* loss-of-function phenotype. In further support of this interpretation, the endocardial over proliferation phenotype that is associated with a loss of *krit1* becomes apparent by 55 hpf (shown in Figure 3—figure supplement 2).

As the reviewers pointed out, there are some phenotypic similarities of the late *krit1* phenotype with the loss of Notch (Beis et al., 2005-this initial description of the Notch inhibited cardiac valve phenotype is now also cited and discussed within the revised manuscript) or loss of *klf2a* phenotypes (Steed et al., 2016). However, in our study, we show that not only *klf2a* mRNA levels are increased (Figure 3) but also that Notch signaling is expressed in a larger domain throughout the endocardium in late loss of *krit1* conditions (Figure 4; Figure 4—figure supplement 3B). Taken together this argues for an activated late Notch and Klf2a signaling that is causing the cardiac valve leaflet morphogenesis defects. This observation raises the interesting point that both a late loss and gain of Klf2a/Notch signaling causes comparable morphological defects in cardiac valve leaflet morphogenesis. We have now also included a short discussion of this point within the revised manuscript.

6) It would be beneficial to provide further support for the requirement of Krit1 for the generation of abluminal cell fates during valvulogenesis. The transient rescue of krit1 mutants allows AVC endocardial cells to differentiate properly at 48 hpf. Alcam levels do appear higher in most AVC cells at 96 hpf (Figure 3, duplicated in Figure 3—figure supplement 4’) but the morphology of the AV valve seems to also include abluminal cells (cardiac jelly cells or mesenchymal-like cells). Repeating the experiment in combination with a Wnt signaling readout (TCF reporter or other) and/or using the inducible hs:Krit1 line in the krit1 mutant background to better control krit1 levels, would strengthen their claim.

We agree with the reviewers on the need to provide additional markers for abluminal versus luminal fates in the rescue experiment. Due to the genetic complexity of setting up this rescue experiment, we have not been able to construct lines that also contain the Wnt reporter strain to show that abluminal fates are indeed absent/ suppressed. Also, the construction of a line for *hs:*Krit1 overexpression in that genetic background was not feasible. Unfortunately, constructing these lines would have exceeded the time allocated for these revisions by several months. Instead, we have now included data on Cdh5, another luminal fate marker. Using this additional marker, we now show that the late loss of Krit1 causes cells in abluminal positions to maintain high levels of Cdh5 while, in WT, Cdh5 is rapidly degraded within abluminal cells (Figure 3—figure supplement 4–C’’’). This finding is further supported by the finding that, in *krit1* mutants, Cdh5 levels are generally increased within endocardium (Figure 3—figure supplement 4).

We have now removed all duplicated figures from the revised manuscript.

7) The model in Figure 5 is somewhat confusing. It might be helpful to show a model for the ccm mutant with blood flow and then when there is no blood flow. Based on the authors' current model, it seems that Notch would be activated throughout the endocardium in the ccm mutant with blood flow since ccm is not present to block klf2a expression. If this is the case, this seems to be in conflict with the current ccm mutant and no blood flow model where the lack of blood flow would lead to no klf2a expression and downstream Notch signaling. Additionally, the authors also speculate "Notch-mediated lateral inhibition" in their model; however, there is insufficient data to support this notion. This should be modified in the model.

We agree with the reviewers that the speculation on “Notch-mediated lateral inhibition” should be removed from the model as shown in the revised Figure 5. Importantly, Klf2a has two modes of induction as indicated in that model figure: 1). the physiological induction by blood flow and 2). a blood-flow independent suppression of expression by CCM proteins. Hence, the loss of CCM proteins causes Klf2a expression (and hence Notch1b expression) also in a blood-flow independent manner.

[Editors' note: further revisions were requested prior to acceptance, as described below.]

The reviewers appreciate the improvements made in your revised manuscript, but they have also raised several issues that remain to be addressed through further revision, as outlined below.1) The Abstract is confusing in its current form, as it is not made clear in which direction many of the genes/pathways mentioned affect outcome. As an example: does the abstract make it clear whether heg1 expression is increased or reduced by low blood flow? The last paragraph of the Introduction section is similarly vague and does not refer to the heg1 results.

The Abstract has been rephrased and clarified to clearly state that *heg1* is positively regulated by blood flow. The revised Abstract now states:

“We find that the expression of *heg1*, which encodes a binding partner of Krit1, is positively regulated by blood flow. In turn, Heg1 stabilizes levels of Krit1 protein and both, Heg1 and Krit1, dampen expression levels of *klf2a*, a major mechanosensitive gene which is a regulator of *notch1b*. Conversely, loss of Krit1 results in increased expression of *klf2a* and *notch1b* throughout the endocardium and also prevents endocardial sprouting which is required for cardiac valve leaflet formation.”

Similarly, the last paragraph of the Introduction section has been clarified:

“We find that expression of *heg1* is positively regulated in response to blood flow and *klf2a*/b. We also explored the role of Heg1 and Krit1 in Klf2-dependent endothelial mechanotransduction during the formation of cardiac valve leaflets. Here, we show that Krit1 dampens levels of *klf2a* and *notch1b* expression. In tune with this finding, loss of Krit1 results in the strong expression of *klf2a* and *notch1b* within the endocardium and prevents the formation of an abluminal population of valve leaflet cells from endocardial cushions.”

2) In the Title, it may be better to mention CCM1/2, since CCM3 is not addressed in this study. Non-CCM experts may not appreciate what "CCM proteins" are.

The title has been changed to explicitly refer to Heg1 and Ccm1/2.

3) From the data shown in the revised manuscript, it is not clear whether heg1 is expressed in the endocardium or the myocardium, although it is stated that it is expressed in the endocardium. Can the evidence for this be clarified?

The endocardial expression has been described by Mably et al., 2003 (Figure 7B). Within the revised manuscript, we now also include an improved Figure 1—figure supplement 1 which shows that *heg1* expression is strongly reduced in *tnnt2a* morphants that lack blood flow. This finding is in tune with a blood flow-dependent regulation of *heg1* within the endocardium.

4) The use of the Notch reporter, in place of the previous Flt reporter, to monitor flow response is appreciated. However, the images in Figure 2 are hard to interpret. Although the ratios measured by the authors (Figure 2) demonstrate the same trend following overexpression of Krit1 and Ccm2, the patterns of expression shown in Figure 2 seem markedly different; hard to interpret. This apparent discrepancy should be addressed. In addition, in light of recent studies showing myocardial Notch reporter activity in developing zebrafish hearts, it is unclear whether the Notch reporter activity shown here (and in Figure 4) is endocardial or myocardial. Can this be clarified?

We apologize for the previous figure which has been misleading with respect to the expression of Notch activity in hs:Ccm2 conditions. Within the revised version of the manuscript, we have now exchanged, within Figure 2 more representative subfigure of the *hs:*Ccm2 condition which is similar to hs:*Krit1* (which is also supported by the quantifications shown in subfigures Figure 2. The strong downregulation of Notch activity upon activation of Krit1 or *ccm2* is also strikingly visible in a new Figure 1. Notably, the expression of this Notch reporter was exclusively endocardial in all of these experimental conditions (Figure 2—figure supplement 1).

5) The authors utilize Cdh5 as a marker for the lumenal side of the AV valve, but they do not cite references for its use as a lumenal marker. If they are introducing this marker here, its characterization as such should be clearly described.

Within the revised manuscript, we have referenced the work of Steed et al., 2016 which has first described Cdh5 as a luminal marker at the AVC valves whereas the protein is lost from the membrane on the abluminal side of the cardiac valve leaflets. This has been shown in Steed et al., 2016, Figure 3 and in the model thereof.

6) In Figure 4—figure supplement 1, the authors use a single marker (Myh6) to define the location of the AVC in krit1 mutants. This marker indicates the edge of the atrium but isn't sufficient to demonstrate whether the AVC is or isn't expanded in the mutant hearts. Can the authors use specific AVC markers to indicate whether the AVC region is expanded in krit1 mutants?

Within the revised version of the manuscript, we have now included a figure showing a whole-mount in situ hybridization against tbx2b, which is a marker of the AVC region (shown in Figure 4—figure supplement 1). We conclude:

“In comparison, the expression of *tbx2b*, another AVC marker gene (Sedletcaia and Evans, 2011), was not expanded in *ccm2^m201^*mutants (Figure 4—figure supplement 1). Hence, the expansion of the *notch1b* expression domain in *krit1* or *ccm2* mutants is not due to a regional expansion of the AVC.”

7) At the beginning of the Discussion section, the authors allude to evidence that Cdh5/VEcadherin plays into this pathway, but such evidence is not reported in this manuscript.

We apologize for this error which has been corrected within the revised version of the manuscript.

8) The new model figure is appreciated, but, in light of what is suggested by Figure 1, it does not address the effect of flow on Heg1 and Krit1 levels. This is confusing – can it be revised to include these elements of the model?

We have now adjusted model Figure 5 to show that the expression of CCM proteins is positively regulated by blood flow and Klf2. We also show that the expression of *klf2a* is affected by blood flow (this activation is dampened by CCM proteins) and that there is another blood flow-independent mechanism of Klf2 activation that is inhibited by CCM proteins in WT. Upon loss of CCM proteins, a blood-flow independent activation of *klf2* expression throughout the endocardium inhibits endocardial sprouting of cardiac valve leaflet cells. Within the revised figure legend, we state:

“Ccm proteins provide a bias in Klf2a and Notch expression levels. Some cells that express lower levels of *klf2a* expression and have a lower Notch activity establish protrusions and migrate into the cardiac jelly (endocardial sprouting). In *ccm* mutants, endocardial cells overexpress *klf2a* and have a higher Notch activity due to a blood flow-independent mechanism. As a consequence, high levels of Notch activity throughout the entire endocardium result in a failure of endocardial sprouting.”